# A citric acid cycle-deficient *Escherichia coli* as an efficient chassis for aerobic fermentations

Hang Zhou [1,2], Yiwen Zhang[1,2], Christopher P. Long [3], Xuesen Xia[4], Yanfen Xue[1], Yanhe Ma [1] ✉, Maciek R. Antoniewicz [3,5] ✉, Yong Tao [1,2] ✉ & Baixue Lin [1,2] ✉

Tricarboxylic acid cycle (TCA cycle) plays an important role for aerobic growth of heterotrophic bacteria. Theoretically, eliminating TCA cycle would decrease carbon dissipation and facilitate chemicals biosynthesis. Here, we construct an *E. coli* strain without a functional TCA cycle that can serve as a versatile chassis for chemicals biosynthesis. We first use adaptive laboratory evolution to recover aerobic growth in minimal medium of TCA cycle-deficient *E. coli*. Inactivation of succinate dehydrogenase is a key event in the evolutionary trajectory. Supply of succinyl-CoA is identified as the growth limiting factor. By replacing endogenous succinyl-CoA dependent enzymes, we obtain an optimized TCA cycle-deficient *E. coli* strain. As a proof of concept, the strain is engineered for high-yield production of four separate products. This work enhances our understanding of the role of the TCA cycle in *E. coli* metabolism and demonstrates the advantages of using TCA cycle-deficient *E. coli* strain for biotechnological applications.

Metabolic engineering of industrial microbes to synthesize value-added products from renewable carbon sources requires high product yields and substrate utilization rates. Microbes have naturally evolved to convert available substrates into cell mass; any attempts to reroute carbon fluxes away from cell growth and towards products of interest disturbs native metabolism and can affect carbon efficiency. For the heterotrophic industrial bacteria *E. coli*, the tricarboxylic acid cycle (TCA cycle) plays multiple roles during aerobic growth. In addition to providing key precursors for biomass synthesis, the conventional TCA cycle completely oxidizes carbon sources such as glucose to $CO_2$, generates reductants (NAD(P)H, $FADH_2$), and produces energy (ATP) primarily through coupling with oxidative phosphorylation. The TCA cycle oxidizes each acetyl-CoA to two $CO_2$. The emission of $CO_2$ from the TCA cycle reduces carbon fluxes directed to products synthesis, thus negatively impacting product yield. Therefore, blocking the TCA cycle and its bypass pathways could decrease carbon dissipation and facilitate chemical biosynthesis in aerobic fermentations.

Strategies that block or shunt TCA cycle flux to raise product yields have thus far encountered growth defects in minimal medium[1–3]. For example, inactivation of α-ketoglutarate dehydrogenase (*ΔsucA*) and glyoxylate shunt (*ΔaceA*)[1–3] resulted in no aerobic growth on glucose. Several strategies have been attempted to overcome this defect. One way was to use an inducible switch of the metabolic flux at the α-ketoglutarate node. By introducing a genetic circuit based on a dual-induction system, conditional interruption of the TCA cycle was used to increase the yield of γ-aminobutyric acid from glucose[4]. Another strategy was to bypass the α-ketoglutarate dehydrogenase reaction using α-ketoglutarate-dependent oxygenase, an enzyme that converts α-ketoglutarate to succinate. As an example, growth of *E. coli ΔaceAΔsucA* strain was enabled on glucose by introducing L-isoleucine dioxygenase[1] or L-proline-4-hydroxylase (P4H)[2,3] with the additional supplementation of isoleucine or proline. However, this latter strategy has several limitations: the corresponding α-ketoglutarate-dependent oxygenases must be active enough in *E. coli*, the co-substrates must be

[1]CAS Key Laboratory of Microbial Physiological and Metabolic Engineering, State Key Laboratory of Microbial Resources, Institute of Microbiology, Chinese Academy of Sciences, 100101 Beijing, China. [2]University of Chinese Academy of Sciences, 100049 Beijing, China. [3]Department of Chemical and Biomolecular Engineering, Metabolic Engineering and Systems Biology Laboratory, University of Delaware, Newark, DE 19716, USA. [4]School of Biology and Biological Engineering, South China University of Technology, Guangzhou 510006, China. [5]Department of Chemical Engineering, University of Michigan, Ann Arbor, MI, USA. ✉e-mail: mayanhe@im.ac.cn; mranton@umich.edu; taoyong@im.ac.cn; linbx@im.ac.cn

efficiently imported or synthesized, and the substrates and products must not be toxic to the cells[1].

Adaptive laboratory evolution (ALE) is a strategy often used to overcome growth defects of engineered strains[5–7]. For example, ALE has been successfully applied to recover aerobic growth of various *E. coli* mutant strains: a strain without phosphotransferase system (PTS)[8], a low NADPH-producing (*ΔzwfΔeddΔeda*) strain[9], and a phosphoglucose isomerase (*pgi*) defective high NADPH-producing strain[10], just to mention a few. In ALE studies, comparative omics analyses and isotope-labeled metabolism flux analysis (¹³C-MFA)[11] are standard procedures for dissecting the genetic basis and metabolic mechanisms for growth recovery. A systematic analysis of an evolved TCA cycle-deficient *E. coli* would be valuable for metabolic engineers who desire to attenuate TCA cycle fluxes in favor of higher product yields.

In previous work[12], a whole-cell biocatalytic process was developed that employed an engineered TCA cycle-deficient (*ΔaceAΔsucA*) *E. coli* strain expressing α-ketoglutarate-dependent deacetoxycephalosporin C synthase (DAOCS) to convert penicillin G to G-7-ADCA. Initially, the TCA cycle-deficient strain was cultured with a rich medium and induced for overexpression of deacetoxycephalosporin C synthase, without the presence of penicillin G, during the cell growth stage. Subsequently, the resting cells, which served as the biocatalyst, were used to produce deacetoxycephalosporin C with penicillin G functioning as the substrate[12]. However, due to the toxic nature of both the substrate and products of DAOCS on *E. coli*, it was impossible to repair the TCA cycle using the DAOCS reaction while cultivating this engineered strain in a glucose minimal medium.

In this work, a TCA cycle-deficient *E. coli* is adaptively evolved to restore aerobic growth in glucose minimal medium. Whole genome sequencing and ¹³C-MFA analysis uncover the mechanisms for growth recovery and identify specific metabolic bottlenecks that are subsequently resolved by targeted engineering of metabolic pathways. We thus construct an efficient *E. coli* strain without a TCA cycle that can be used for production of chemicals. As a proof of concept, we use the optimized *E. coli* strain for high-yield biosynthesis of four separate products in aerobic fermentations, thus demonstrating the potential of using an *E. coli* strain with a severed TCA cycle for biotechnological applications.

## Results
### Growth recovery of TCA cycle-deficient *E. coli* after ALE
Adaptive laboratory evolution was performed on the TCA cycle-deficient *E. coli* strain dTCA (BW25113 *ΔaceAΔsucAΔgadAΔgadBΔpoxB::acs*, Supplementary Table 1) to restore aerobic growth in glucose minimal medium. The TCA cycle-deficient strain was precultivated overnight in Luria-Bertani medium. For the first and second transfers, 5 mL culture was transferred into 50 mL glucose minimal medium supplemented with 5 mL and 0.5 mL Luria-Bertani respectively. For the following serial passages, 0.5 mL culture was transferred into 50 mL glucose minimal medium without supplements. The dTCA strain lost a key gene of TCA cycle, α-ketoglutarate dehydrogenase (*ΔsucA*), which encodes for the enzyme responsible for converting α-ketoglutarate to succinyl-CoA, as well as genes for two bypass pathways, the glyoxylate shunt (*ΔaceA*) and γ-aminobutyric acid shunt (*ΔgadAΔgadB*). In order to monitor the acetate (acetyl-CoA) pool and determine if the TCA cycle in the dTCA strain remained blocked during evolution, the pyruvate oxidase gene *poxB* was replaced with the acetyl-CoA synthetase gene *acs* (*ΔpoxB::acs*). In this situation, acetate formed from acetyl-CoA was converted back to the acetyl-CoA pool.

Adaptive laboratory evolution was conducted by continuous aerobic culture in glucose minimal medium. After 48 days of evolution (~230 generations), cells regained the ability to grow aerobically in glucose minimal medium. The specific growth rates of two evolutionary endpoint strains (μ = 0.61 ± 0.02 h⁻¹ for dTCA-E1, μ=0.59 ± 0.02 h⁻¹ for dTCA-E2) were similar to that of a control strain

iTCA with an intact TCA cycle (BW25113 *ΔpoxB::acs*, μ=0.64 ± 0.02 h⁻¹) (Fig. 1a). The acetate yields of dTCA-E1 (0.82 ± 0.06 mol/mol) and dTCA-E2 (0.79 ± 0.12 mol/mol) during log-phase were significantly higher than that of iTCA strain (0.38 ± 0.01 mol/mol, Fig. 1b). When ALE was repeated, similar results were obtained. Two evolutionary endpoint strains had comparable growth rates (0.62 ± 0.04 h⁻¹ for dTCA-E3, 0.63 ± 0.04 h⁻¹ for dTCA-E4) and increased acetate yields (0.97 ± 0.03 mol/mol for dTCA-E3, 0.92 ± 0.04 mol/mol for dTCA-E4) (Fig. 1a, b). The biomass yields of all evolutionary endpoint strains were significantly lower than that of the iTCA strain (Fig. 1c), while the glucose uptake rates of those four evolved strains were significantly higher than that of iTCA strain (Fig. 1d).

### Mutations in genes encoding TCA cycle enzymes in evolved *E. coli*
Whole genome sequencing (https://www.ncbi.nlm.nih.gov/sra/PRJNA967749) and mutations analysis (the result of genomic mutations analysis was in Supplementary Data 1–6) were performed on the evolutionary endpoint strains and the unevolved dTCA strain to uncover the mechanisms behind growth recovery. The analysis of gene mutations in the evolved strains revealed common mutations in genes encoding enzymes in the TCA cycle (Fig. 1e, Supplementary Data 6). Specifically, genes encoding for succinate dehydrogenase subunit A (*sdhA*) and citrate synthase (*gltA*) were mutated in all four evolutionary endpoint strains. To investigate how these mutations affected enzyme activity, the evolved strains were cultivated in glucose medium to log-phase and enzymatic activity of succinate dehydrogenase and citrate synthase were measured. No significant succinate dehydrogenase activity was detected in the lysate of the four evolved strains containing succinate dehydrogenase mutations (Fig. 1f). The activities of GltA(T109P) (0.019 ± 0.002 U/mg total protein), GltA(L10I, T109P) (0.019 ± 0.001 U/mg total protein), GltA(H157Y) (0.040 ± 0.001 U/mg total protein) and GltA(I375T) (0.019 ± 0.000 U/mg total protein) were significantly attenuated compared to the control strain iTCA (0.054 ± 0.003 U/mg total protein) (Fig. 1g). These results suggested that inactivation of succinate dehydrogenase and attenuation of citrate synthase were critical for growth recovery of TCA cycle defected *E. coli*.

To further test this hypothesis, *sdhA* was deleted in the unevolved TCA cycle-deficient starting strain dTCA and *gltA* was replaced with the identified *gltA* mutants (GltA(T109P), GltA(L10I, T109P), GltA(H157Y) and GltA(I375T)). As shown in Fig. 1h, deletion of *sdhA* was sufficient to restore aerobic growth (μ = 0.35 ± 0.01 h⁻¹, Supplementary Table 2), which was consistent with the previous work[13]. While mutations of *gltA* alone could not restore aerobic growth of dTCA strain (Fig. 1h, Supplementary Table 2), they did significantly improve aerobic growth rate of the strain dTCA *ΔsdhA* (Fig. 1i, Supplementary Table 2). These results suggested that succinate dehydrogenase inactivation was essential for the growth recovery of TCA cycle defected *E. coli*, while the attenuation of citrate synthase was beneficial but not necessary.

### Metabolism rewiring in evolved TCA cycle-deficient *E. coli*
To elucidate the metabolic mechanisms for growth recovery of evolved TCA cycle-deficient *E. coli*, ¹³C-metabolism flux analysis (¹³C-MFA) was performed on the evolved dTCA-E1 strain and iTCA strain, which had a complete TCA cycle (Data of mass isotopomer distributions were listed in Supplementary Data 7, Data of MFA were shown in Supplementary Data 8). The ¹³C-MFA results uncovered extensive metabolic flux rewiring all throughout central carbon metabolism (Fig. 2a), and provided additional support for the genome sequencing results and in vitro enzyme activity measurements. In the evolved dTCA-E1 strain, no metabolic flux was detected from α-ketoglutarate to succinyl-CoA (0 ± 0), consistent with the knockout of *sucA* gene; and from succinate to fumarate (0 ± 0), consistent with the presumed

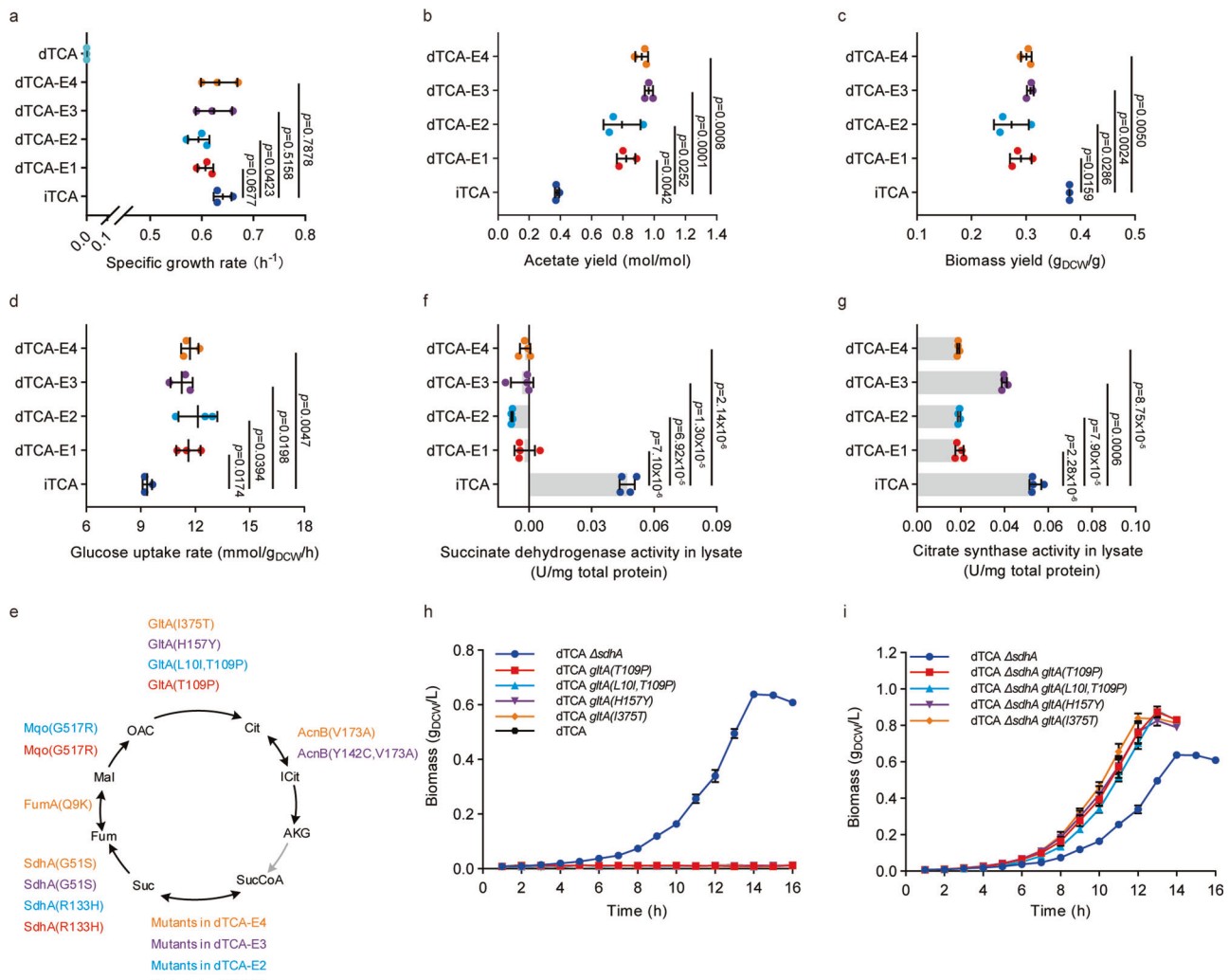

**Fig. 1 | The TCA cycle-defective *Escherichia coli* strain dTCA showed aerobic growth recovery in glucose minimal medium when a succinate dehydrogenase mutation was introduced. a** Growth rates of evolved TCA cycle-deficient strains comparing to iTCA strain. Values are means ± SD (*n* = 3 biological replicates). **b** Acetate yields of evolved TCA cycle-deficient strains comparing to iTCA strain. Values are means ± SD (*n* = 3 biological replicates). **c** Biomass yields of evolved TCA cycle-deficient strains comparing to iTCA strain. Values are means ± SD (*n* = 3 biological replicates). **d** Glucose uptake rates of evolved TCA cycle-deficient strains comparing to iTCA strain. Values are means ± SD (*n* = 3 biological replicates). **e** Mutated genes encoding TCA cycle enzymes in two groups of independent ALE experiments. GltA: citrate synthase; AcnB: aconitate hydratase B; SdhA: succinate dehydrogenase subunit A; FumA: fumarase A; Mqo: malate: quinone

oxidoreductase. **f** Enzyme activities of mutated succinate dehydrogenases in evolved strains. Values are means ± SD (*n* = 4 biological replicates). **g** Enzyme activities of mutated citrate synthases in evolved strains. Values are means ± SD (*n* = 4 biological replicates). **h** Aerobic growth of unevolved dTCA strain after introducing succinate dehydrogenase or citrate synthase mutations. Values are means ± SD (*n* = 3 biological replicates). **i** Aerobic growth of unevolved dTCA *ΔsdhA* strain after introducing citrate synthase mutations. Values are means ± SD (*n* = 3 biological replicates). All above *P* values were calculated by unpaired two-tailed *t* test. dTCA: BW25113 *ΔaceAΔsucAΔgadAΔgadBΔpoxB::acs*; iTCA: BW25113 *ΔpoxB::acs*; dTCA-E1, dTCA-E2, dTCA-E3, dTCA-E4: evolved dTCA strains. Source data are provided as a Source Data file.

inactivation of SdhA (R133H) in dTCA-E1 strain (Fig. 1f). Moreover, citrate synthase flux was significantly reduced in dTCA-E1 (7 ± 0) compared to iTCA strain (27 ± 1), consistent with citrate synthase activity attenuation (Fig. 1g). In the dTCA-E1 strain, succinyl-CoA was regenerated exclusively from succinate, while in the iTCA strain all of succinyl-CoA was derived from the decarboxylation of α-ketoglutarate in the TCA cycle. As SdhA was inactivated, succinate pool in dTCA-E1 might be supplemented from fumarate by L-aspartate oxidase (NadB)[14,15] or fumarate reductase (Frd)[16] (Supplementary Fig. 1). The low regeneration flux of succinyl-CoA in dTCA-E1 (2 ± 0) compared to iTCA (14 ± 1) further suggested that this step may be a metabolic bottleneck for cell growth given that succinyl-CoA is required for the biosynthesis of several amino acids in *E. coli*[17].

In the dTCA-E1 strain, only 2.2% of carbon was converted into $CO_2$ via the TCA cycle, whereas in the iTCA strain, this percentage was much

higher at 11.7% (Fig. 2b, Supplementary Data 8). Notably, $CO_2$ production in the dTCA-E1 strain was primarily associated with the synthesis of α-ketoglutarate. Most of the $CO_2$ released by dTCA-E1 was generated during the formation of acetate from pyruvate (Supplementary Data 8).

In addition to altered TCA cycle flux patterns, significant rewiring of upper metabolism was observed in the evolved strain. The oxidative pentose phosphate pathway (oxPPP) flux was significantly upregulated in dTCA-E1 compared to iTCA (37 ± 0 vs. 24 ± 1) (Fig. 2a). As a result, more than 85% of NADPH required for cell growth was produced via oxPPP in dTCA-E1 compared to less than 40% in iTCA (Fig. 2c, Supplementary Data 8). Remarkably, the transhydrogenase flux was negligible in dTCA-E1 strain (3 ± 3), whereas in iTCA the transhydrogenase flux was a major contributor to biosynthetic NADPH production (Fig. 2c, Supplementary Data 8).

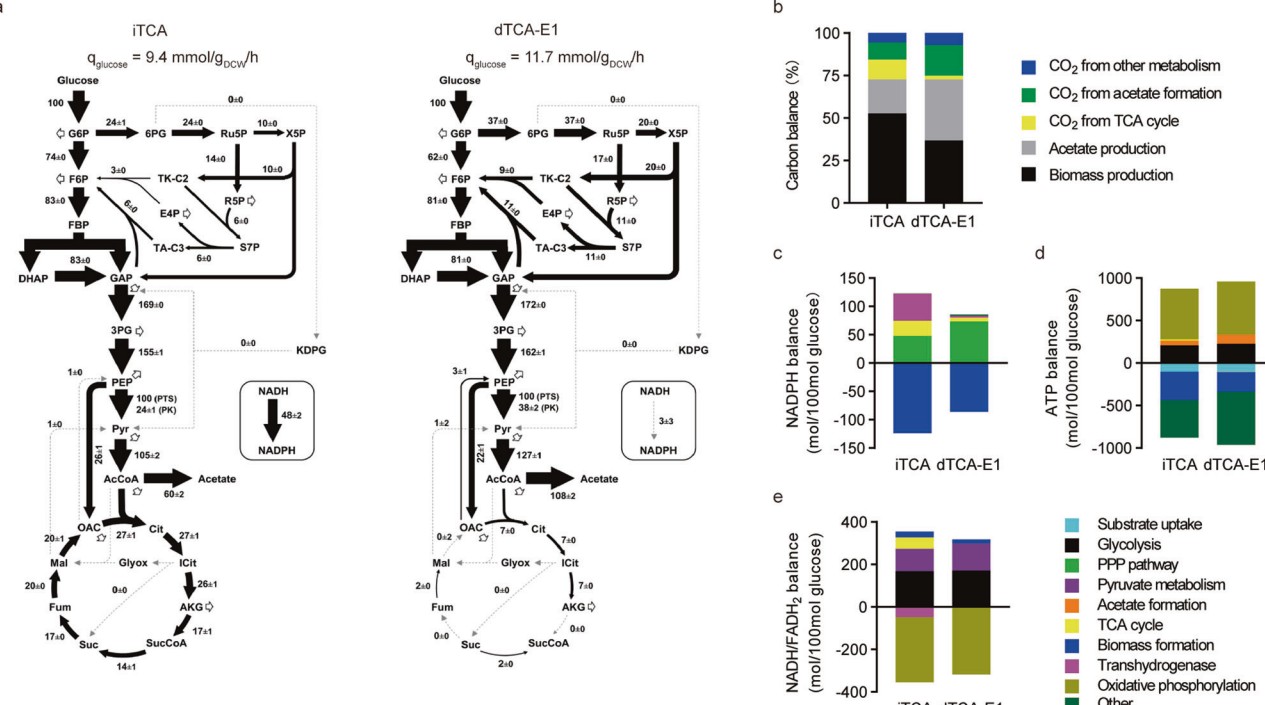

**Fig. 2 | $^{13}$C-metabolic flux analysis reveals rewired central carbon metabolism to counter a TCA cycle defect. a** Central carbon metabolism flux distributions (normalized to 100 mol glucose uptake) of the evolved strain dTCA-E1 and the unevolved strain iTCA. Values are means ± SD ($n$ = 3 biological replicates). **b** Carbon balance analysis of dTCA-E1 and iTCA. Values are means ± SD ($n$ = 3 biological replicates). **c** NADPH metabolism balance analysis of dTCA-E1 and iTCA. Values are means ± SD ($n$ = 3 biological replicates). **d** ATP metabolism balance analysis of dTCA-E1 and iTCA. Values are means ± SD ($n$ = 3 biological replicates).

**e** NADH/FADH$_2$ metabolism balance analysis of dTCA-E1 and iTCA. Values are means ± SD ($n$ = 3 biological replicates). The co-factor metabolism balance analysis was shown as the normalized production (positive value) and consumption (negative value) per 100 mol of glucose consumed. dTCA-E1: evolved TCA cycle deficient dTCA strain (Fig. 1, Supplementary Table 1); iTCA: strain BW25113 $\Delta pox$-$B$::acs with an intact TCA cycle (Fig. 1, Supplementary Table 1). Source data are provided as a Source Data file.

The glycolytic flux to acetyl-CoA was significantly higher in dTCA-E1 than in iTCA (127 ± 1 vs. 105 ± 2). This increased glycolysis flux, combined with reduced citrate synthase flux, produced a metabolic imbalance that was manifested as increased acetate secretion in dTCA-E1 compared to iTCA (108 ± 1 vs. 60 ± 2). The remodeling of central carbon metabolism also affected energy generation. Compared to strain iTCA, dTCA-E1 obtained more ATP from substrate level phosphorylation and acetate production (Fig. 2d, Supplementary Data 8) and showed higher overall ATP metabolism efficiency. The TCA cycle of dTCA-E1 neither directly generate ATP (Fig. 2d) nor produce NADH/FADH$_2$ (Fig. 2e) for coupling with oxidative phosphorylation to produce ATP. Since the absolute glucose uptake rate of dTCA-E1 strain (11.7 mmol/g$_{DCW}$/h) was significantly higher than that of iTCA (9.4 mmol/g$_{DCW}$/h) (Fig. 2a, Supplementary Table 3), we determined that the absolute ATP production rate was nearly 40% higher in dTCA-E1 strain (113 mmol/g$_{DCW}$/h) than in iTCA strain (83 mmol/g$_{DCW}$/h) (Supplementary Fig. 2c, Supplementary Data 8). These results suggested that the TCA cycle was not important for energy generation in dTCA-E1, in which ATP metabolism was efficiently rebalanced through coordinated remodeling of central carbon metabolism.

### Growth recovery by eliminating the requirement for succinyl-CoA

Based to the results of the metabolic mechanism and genetic basis for aerobic growth recovery, we hypothesized that limitations in the supply of succinyl-CoA could be the cause for the inability of TCA cycle-deficient *E. coli* to grow in glucose minimal medium. Succinyl-CoA is involved in the biosynthesis of methionine, meso-2,6-Diaminopimelate (meso-DAP) and lysine, and participates in tetrapyrroles and heme synthesis[17]. When the TCA cycle was blocked at the node of

α-ketoglutarate dehydrogenase, succinyl-CoA could only be synthesized from succinate. We reasoned that availability of succinyl-CoA would be negatively correlated with succinate dehydrogenase activity that consumes succinate and thus there would be a strong selective pressure for inactivation of succinate dehydrogenase during adaptive evolution. Succinate dehydrogenase mutations were found in all 30 randomly isolated clones (Supplementary Tables 4–6) when we sequenced the succinate dehydrogenase operon in dTCA or BW25113 $\Delta aceA\Delta sucA$ clones that survived on glucose M9 minimal medium plate. This finding indicated that succinate dehydrogenase mutation was seemingly inevitable. However, *E. coli* with defective succinate dehydrogenase can't be used to produce chemicals like G-7-ADCA, as it requires the TCA cycle to regenerate α-ketoglutarate from succinate. Hence, we explored an alternative strategy to circumvent the succinyl-CoA requirement. As heme and tetrapyrroles can be synthesized from glumatyl-tRNA instead of succinyl-CoA in *E. coli*[18], we hypothesized that the causal bottleneck for growth of TCA cycle-deficient *E. coli* was the biosynthesis of methionine, meso-DAP and lysine due to insufficient succinyl-CoA supply. To test this hypothesis, strain BW25113 $\Delta aceA\Delta sucA$ was grown in glucose minimal medium supplemented with either succinate (2 mmol/L) or an amino acid mixture (1 mmol/L each of meso-2,6-diaminoheptanedioate, lysine and methionine). In both media, strain BW25113 $\Delta aceA\Delta sucA$ grew well, which was faster than strain BW25113 $\Delta aceA\Delta sucA\Delta sdhA$ in M9 minimal medium (Fig. 3a, Supplementary Table 7). Sequencing of the *sdhCDAB* gene locus in BW25113 $\Delta aceA\Delta sucA$ strain during aerobic growth with supplements revealed no mutations of succinate dehydrogenase. Additionally, the intracellular succinyl-CoA level in glucose-grown BW25113 $\Delta aceA\Delta sucA$ strain supplemented with amino acid mixture (1 mmol/L each of DAP, lysine and methionine) was measured as 208.21 ± 37.82 nmol/g$_{DCW}$,

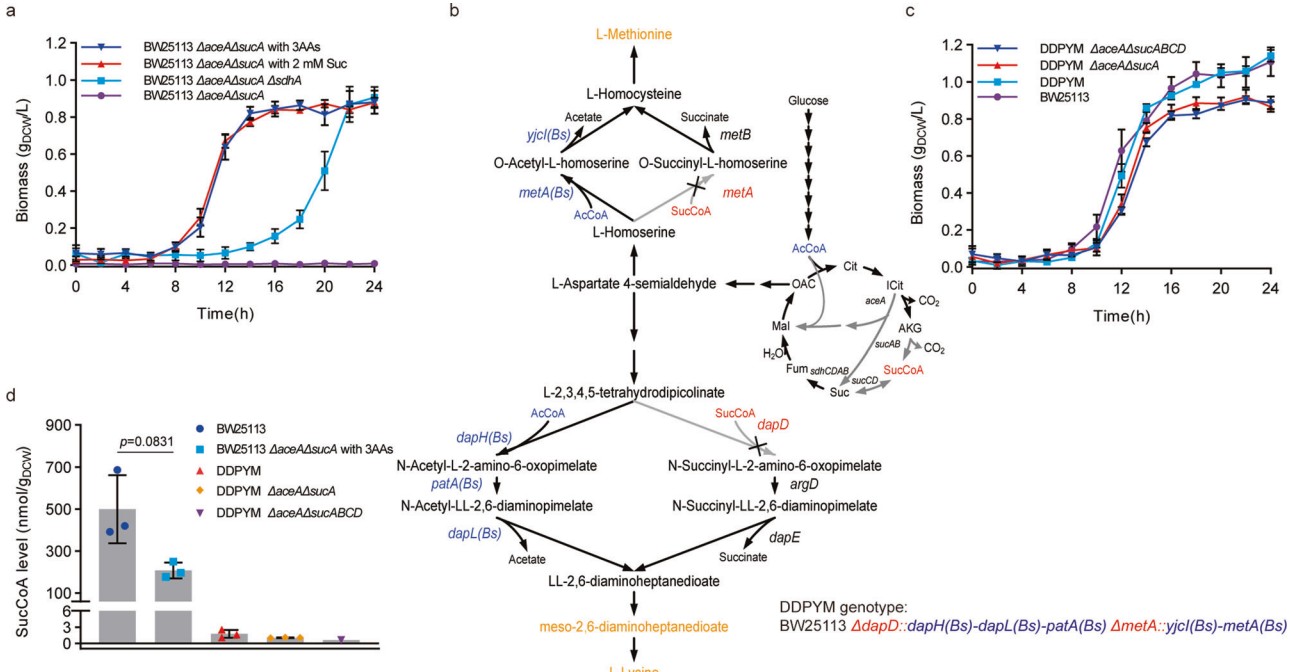

**Fig. 3 | Strategies to recover aerobic growth of TCA cycle-deficient _E. coli_ in glucose minimal medium. a** Aerobic growth of TCA cycle-deficient _E. coli_ BW25113 _ΔaceAΔsucA_ with supplements. Values are means ± SD (_n_ = 6 biological replicates). Aerobic growth was conducted in 96-well plates shaking under 800 rpm and 37 °C. Suc: 2 mmol/L succinate; 3AAs: 1 mmol/L for each of meso-2,6-diaminoheptanedioate (meso-DAP), lysine and methionine. **b** Schematic diagram of reprogramming _E. coli_ to bypass succinyl-CoA requirement. **c** Aerobic growth of TCA cycle-deficient _E. coli_ DDPYM strains without supplements. Values

are means ± SD (_n_ = 6 biological replicates). Aerobic growth was conducted in 96-well plates shaking under 800 rpm and 37 °C. **d** Intracellular succinyl-CoA level in TCA cycle-deficient _E. coli_ strains comparing to strain BW25113. Values are means ± SD (_n_ = 3 biological replicates). Aerobic growth was conducted with 50 mL culture in 250 mL flasks shaking under 200 rpm and 37 °C. 3AAs: 1 mmol/L for each of meso-2,6-diaminoheptanedioate (meso-DAP), lysine and methionine. All above _P_ values were calculated by unpaired two-tailed _t_ test. Source data are provided as a Source Data file.

which was significantly lower than glucose-grown BW25113 (499.28 ± 162.47 nmol/g$_{DCW}$) (Fig. 3d, the intracellular succinyl-CoA levels were in Supplementary Data 9). This result provided further evidence for our hypothesis that insufficient availability of succinyl-CoA was the cause for growth arrest in TCA cycle-deficient _E. coli_.

Some bacteria have evolved alternative metabolic pathways for the biosynthesis of methionine, meso-DAP and lysine that do not require succinyl-CoA (Supplementary Fig. 3)[19–21]. For example, in _Bacillus subtilis_, lysine and methionine are synthesized through acetyl-CoA dependent synthesis pathways. We hypothesized that by introducing acetyl-CoA dependent enzymes from _B. subtilis_ into _E. coli_, the aerobic growth of TCA cycle-deficient _E. coli_ might be recovered in glucose minimum medium. To test this, the acetyl-CoA dependent pathways for biosynthesis of methionine, meso-DAP and lysine from _B. subtilis_ were introduced into _E. coli_ BW25113 resulting in _E. coli_ DDPYM strain (BW25113 _ΔdapD::dapH(Bs)-dapL(Bs)-patA(Bs) ΔmetA::yjcI(Bs)-metA(Bs)_) (Fig. 3b, Supplementary Table 1). Strain DDPYM had a similar growth rate (0.54 ± 0.05 h⁻¹) as wild-type _E. coli_ BW25113 (0.54 ± 0.06 h⁻¹) in glucose minimal medium (Fig. 3c, Supplementary Table 7). Next, genes involved in TCA cycle and glyoxylate shunt (_sucABCD_ and _aceA_) were knocked-out in DDPYM resulting in strains DDPYM _ΔaceAΔsucA_ and DDPYM _ΔaceAΔsucABCD_. Known succinyl-CoA synthesis pathways in DDPYM _ΔaceAΔsucABCD_ were whole removed (_ΔsucABCD_). The specific growth rates of DDPYM _ΔaceAΔsucA_ and DDPYM _ΔaceAΔsucABCD_ were 0.51 ± 0.05 h⁻¹ and 0.52 ± 0.04 h⁻¹, respectively (Fig. 3c, Supplementary Table 7), which were significantly higher than that of strain BW25113 _ΔaceAΔsucAΔsdhA_ (0.32 ± 0.04 h⁻¹) and similar to DDPYM and BW25113 (Fig. 3c, Supplementary Table 7). The levels of succinyl-CoA in strains DDPYM and DDPYM _ΔaceAΔsucA_ were determined to be 1.75 ± 0.73 nmol/g$_{DCW}$ and 1.04 ± 0.06 nmol/ g$_{DCW}$, respectively (Fig. 3d, Supplementary Data 9). While, the succinyl-

CoA level in strain DDPYM _ΔaceAΔsucABCD_ (strain ZH40) was too low to be accurately measured in biological replicate tests (Fig. 3d, Supplementary Data 9). These results confirm that growth arrest observed in TCA cycle-deficient _E. coli_ strains was indeed the results of limitations in succinyl-CoA supply and not energy supply. In short, by introducing heterogeneous acetyl-CoA dependent pathways for biosynthesis of methionine, meso-DAP and lysine, _E. coli_ could successfully grow with a severed TCA cycle (_ΔaceAΔsucA_ or _ΔaceAΔsucABCD_), which was enabled by eliminating the requirement for succinyl-CoA in amino acid biosynthesis pathways[17,22].

## TCA cycle-deficient _E. coli_ as an efficient aerobic chassis

To demonstrate that TCA cycle-deficient DDPYM can serve as a versatile and efficient chassis strain for aerobic fermentations without requirement of succinyl-CoA for amino acid (methionine, meso-DAP and lysine) biosynthesis, strain ZH40 (DDPYM _ΔaceAΔsucABCD_) without succinyl-CoA synthesis pathways (_sucAB_ and _sucCD_)[22] was constructed and engineered for biosynthesis of several chemicals (Fig. 4a). In previous work, we used a TCA cycle-deficient _E. coli_ (_ΔaceAΔsucA_) to convert penicillin G to G-7-ADCA by overexpressing deacetoxycephalosporin C synthase (DAOCS)[12]. As penicillin G and G-7-ADCA was toxic to _E. coli_, the defective TCA cycle could not be repaired with α-ketoglutarate-dependent DAOCS to support cell growth in glucose minimal medium. Here, DAOCS was overexpressed in strain ZH40, which can grow aerobically on glucose minimal medium. Protein expression of DAOCS in ZH40 was similar to that in BW25113 and DDPYM background strains with an intact TCA cycle (Fig. 4b). Notably, G-7-ADCA production in ZH40 host strain (7.02 ± 0.27 mmol/L) was more than 3-fold higher compared to BW25113 (2.20 ± 0.31 mmol/L) and DDPYM background strains (1.45 ± 0.05 mmol/L) (Fig. 4c).

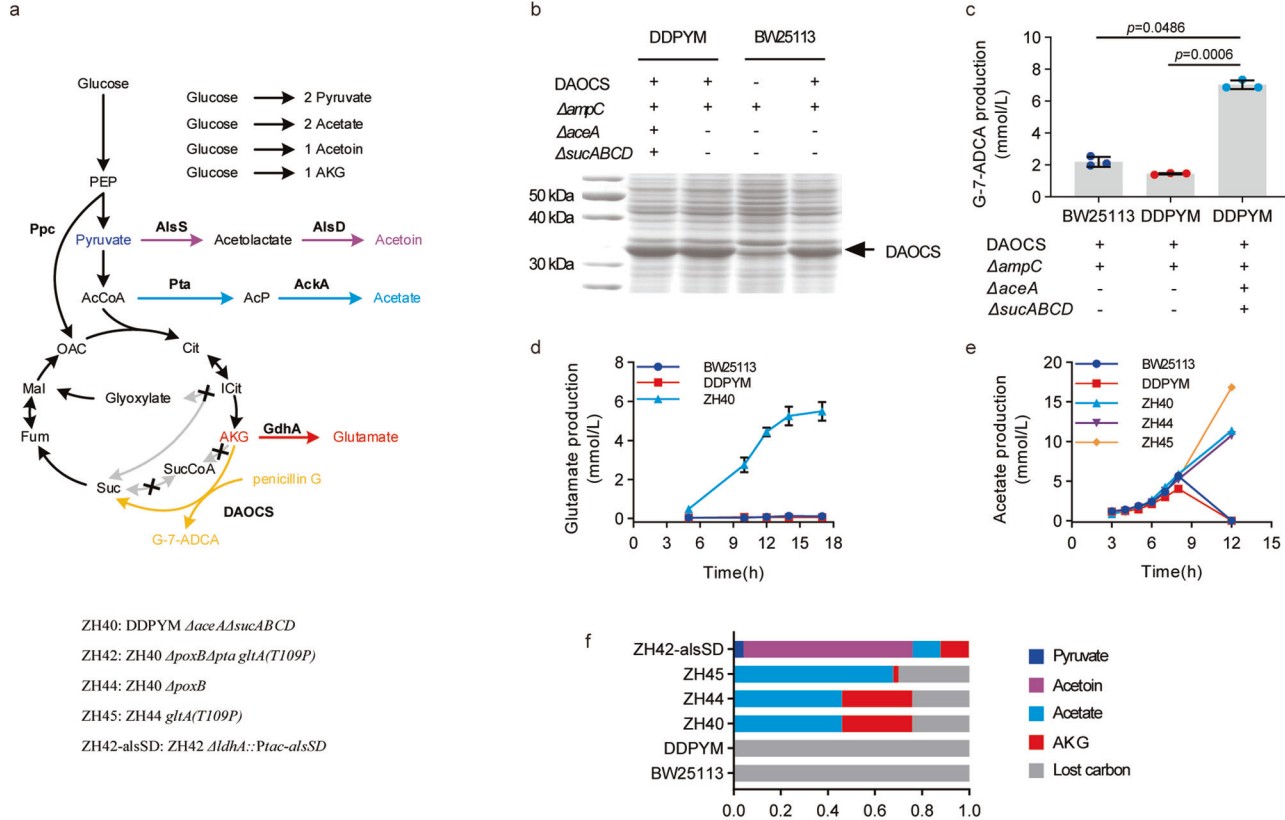

**Fig. 4 | High yield biosynthesis of chemicals using TCA cycle-deficient *E. coli* strains. a** Overview of chemicals biosynthesis using engineered DDPYM strains. Ppc: phosphoenolpyruvate carboxylase; GdhA: glutamate dehydrogenase; AlsS: alpha-acetolactate synthase; AlsD: alpha-acetolactate decarboxylase; AckA: acetate kinase A; Pta: phosphate acetyltransferase; DAOCS: deacetoxycephalosporin C synthase. **b** Overexpression of deacetoxycephalosporin C synthase (DAOCS) using TCA cycle-deficient ZH40 strain as chasiss. Inducible expression of DAOCS was conducted via 6 h cultivation after adding 0.5 mmol/L IPTG at 0.5 OD$_{600}$. **c** Production of G-7-ADCA using TCA cycle-deficient ZH40 strain as chasiss. Values are means ± SD ($n$ = 3 biological replicates). **d** Production of glutamate in TCA cycle-deficient chassis ZH40 with plasmid pSC2s-gdhA-ppc. Values are means ± SD ($n$ = 3 biological replicates). **e** Production of acetate in TCA cycle-deficient chassis strains comparing to BW25113 and DDPYM. Values are means ± SD ($n$ = 3 biological replicates). **f** Fraction of carbon flux from glucose to metabolites during whole-cell catalysis. Values are means ± SD ($n$ = 3 biological replicates). DDPYM: BW25113 Δ*dapD::dapH(Bs)-dapL(Bs)-patA(Bs)* Δ*metA::yjcI(Bs)-metA(Bs)*. Detailed information on all strains is listed in Supplementary Table 1. All above *P* values were calculated by unpaired two-tailed *t* test. Source data are provided as a Source Data file.

Next, strain ZH40 was engineered to produce glutamate, one of the twelve top value-added chemicals from biomass. By over-expressing phosphoenolpyruvate carboxylase (*ppc*) and glutamate dehydrogenase (*gdhA*) (Fig. 4a), 5 mmol/L glutamate was obtained in 17 h during aerobic growth on 20 mmol/L glucose (Fig. 4d, Supplementary Fig. 4a) by strain ZH40-gdhA-ppc, while no detectable glutamate was observed in strains with an intact TCA cycle BW25113-gdhA-ppc and DDPYM-gdhA-ppc strains (Fig. 4d, Supplementary Fig. 4a). In the whole-cell biocatalysis, little amount of AKG were detected in BW25113 and DDPYM (Supplementary Fig. 5a, b), while strain ZH40 (DDPYM Δ*aceA*Δ*sucABCD*) with defective TCA cycle produced 14.93 ± 0.50 mmol/L AKG after complete consumption of 50 mmol/L glucose (Supplementary Fig. 5c). These results demonstrated the advantage of the defective TCA cycle on AKG accumulation and the TCA-cycle defective strain held potential in production of chemicals derived from AKG.

To demonstrate that strain ZH40 can also be used for efficient production of acetyl-CoA derived products, pyruvate oxidase gene (*poxB*) was knocked-out from strain ZH40, resulting in strain ZH44 (DDPYM Δ*aceA*Δ*sucABCD*Δ*poxB*) which could only generate acetate from acetyl-CoA. Strains ZH40 and ZH44 produced acetate at a higher yield and rate than *E. coli* strains with an intact TCA cycle (Fig. 4e, Supplementary Table 8), while maintaining a similar growth rate (0.63 ± 0.01 h$^{-1}$ and 0.64 ± 0.00 h$^{-1}$, respectively) as BW25113

(0.66 ± 0.01 h$^{-1}$) and DDPYM (0.63 ± 0.00 h$^{-1}$) (Supplementary Fig. 4b, Supplementary Table 8). When citrate synthase of ZH44 was mutated as GltA(T109P) to reduce flux of acetyl-CoA into TCA cycle, the growth rate, biomass yield, acetate yield and acetate production rate of the resulting strain ZH45 (ZH44 *gltA(T109P)*) were increased during exponential growth phase (Fig. 4e, Supplementary Fig. 4b, Supplementary Table 8). During whole-cell biocatalysis, acetate yield increased further to 0.96 ± 0.01 mol/mol in ZH40 and ZH44 (Supplementary Fig. 5c, d, Figs. 4f), and 1.42 ± 0.04 mol/mol in ZH45 (Supplementary Fig. 5e, Fig. 4f), while no acetate was accumulated in cultures of *E. coli* strains with an intact TCA cycle (Supplementary Fig. 5a, b, Fig. 4f). These results demonstrated the benefit of the defective TCA cycle for production of chemicals from acetyl-CoA.

In whole-cell catalysis stage, the resting cells did not grow. The yields of metabolites from glucose in whole-cell catalysis stage (Supplementary Fig. 5) were used to analyze the distribution of metabolic flow (Fig. 4f). To calculate the fraction of carbon fluxes from glucose to metabolites (Fig. 4f), the measured yields of metabolites were divided by the theoretical yields of their respective pathways, with CO$_2$ and any unmeasured metabolites considered as lost carbon. In the strains with complete TCA cycle, BW25113 and DDPYM, the TCA cycle decarboxylation caused complete carbon loss from glucose (Fig. 4f, Supplementary Fig. 5a, b). However, the TCA cycle-deficient strains ZH40, ZH44, and ZH45 showed a significant reduction in carbon loss, with

over 70% of the carbon flux diverted to acetate and α-ketoglutarate (AKG) (Fig. 4f, Supplementary Fig. 5c–e). Mutation of citrate synthase in strain ZH45 leads to a decrease in AKG flux and an increase in acetate flux compared to strain ZH44 (Fig. 4f, Supplementary Fig. 5d, e). However, these TCA cycle defective strains ZH40, ZH44, and ZH45 also presented above 20% carbon loss. Citrate synthase mutation caused a modest increase in lost carbon in strain ZH45 (26.54 ± 2.17%) comparing to strain ZH44 (20.44 ± 0.92%) (Fig. 4f). We hypothesized that there might exist carbon loss from acetyl-CoA or pyruvate in strain ZH45, which might be decreased if carbon flux was directed from pyruvate to pyruvate-derived chemical such as acetoin. Acetoin can be synthesized from pyruvate using acetolactate synthase (AlsS) and acetolactate decarboxylase (AlsD) without production or consumption of reductants or energy (Fig. 4a). Genes *alsS* and *alsD* from *Bacillus subtilis* were introduced into strain ZH42, resulting in strain ZH42-alsSD (Supplementary Table 1). In whole-cell catalysis, strain ZH42-alsSD generated 35.98 ± 0.57 mmol/L acetoin from 50 mmol/L glucose, which was 72% of the theoretical maximal yield from glucose (Supplementary Fig. 5f). Only 0.17 ± 0.67% of the carbon in ZH42-alsSD was lost from glucose, and most of carbon flowed to metabolites including pyruvate (4.14 ± 0.94%), acetoin (71.95 ± 1.13%), acetate (11.65 ± 0.87%), and α-ketoglutarate (12.09 ± 0.35%) (Fig. 4f). These results demonstrated the significant potential of TCA cycle-deficient DDPYM strains for minimizing carbon dissipation and facilitating chemicals production.

## Discussion

The conventional TCA cycle in heterotrophic microbes has been frequently modified to improve chemicals production in aerobic fermentation[1–4,12,23,24]. While some microorganisms such as cyanobacteria[25], methylotrophic bacteria[26,27] and *Mycobacterium tuberculosis*[28,29] have been found to have an incomplete TCA cycle. However, it was found that a break in the TCA cycle seriously affects aerobic growth of *E. coli* in glucose minimal medium[1–3]. In this work, we successfully reprogramed metabolism of *E. coli* without a complete TCA cycle to grow well on glucose and convert glucose into products at high yields.

Adaptive laboratory evolution was used to gain a better understanding of the underlying mechanisms causing the growth defect observed when TCA cycle was severed at the α-ketoglutarate dehydrogenase step in *E. coli*. It was discovered that the limiting factor for growth was not energy limitation but insufficient supply of succinyl-CoA resulting in the inability to produce three amino acids, methionine, meso-DAP and lysine. Succinyl-CoA can only be generated from succinate in *E. coli* if the α-ketoglutarate dehydrogenase reaction was blocked (ΔsucA). When the glyoxylate cycle was also blocked by inactivation of isocitrate lyase (ΔaceA), the *E. coli* with defective TCA-cycle (ΔaceAΔsucA) lacked sufficient succinate to maintain the succinyl-CoA pool. Although succinate might be supplemented from fumarate by L-aspartate oxidase (NadB)[14,15] or fumarate reductase (Frd)[16] (Supplementary Fig. 1), the amount of succinate was limited. Due to succinate being rapidly oxidized by succinate dehydrogenase, TCA cycle defected (ΔaceAΔsucA) *E. coli* could not maintain a sufficient succinyl-CoA pool without enough intracellular succinate, unless its succinate dehydrogenase was mutated. Upon transfer to glucose minimal medium plates from rich medium, those survived dTCA and BW25113 ΔaceAΔsucA clones were found to have succinate dehydrogenase mutations (Supplementary Tables 4–6). This work provides evidence that succinate dehydrogenase mutation is necessary and unavoidable for recovering aerobic growth of TCA-cycle defective *E. coli* (ΔaceAΔsucA) in glucose minimal medium.

However, succinate dehydrogenase mutation was inapplicable to the production of chemicals such as G-7-ADCA, which required a reconstituted TCA cycle to regenerate α-ketoglutarate from succinate[12]. Therefore, we attempted to circumvent succinyl-CoA and

construct a more versatile chassis. It is a challenge to bypass succinyl-CoA requirement, for succinyl-CoA has always been considered as one of the twelve essential precursor metabolites for cell growth[17]. This work showed that growth could be recovered by bypassing the requirement for succinyl-CoA in amino acids synthesis through expression of heterologous acetyl-CoA dependent enzymes. Succinyl-CoA synthesis pathways genes *sucAB* and *sucCD* were knocked out in TCA cycle-deficient *E. coli* strain ZH40, which still retained the ability to provide essential precursor metabolites and energy for aerobic growth (Fig. 3c, Supplementary Fig. 4b) and over expression of protein (Fig. 4b) while avoiding unnecessary oxidization of acetyl-CoA to $CO_2$ in the TCA cycle (Fig. 4e, f, Supplementary Fig. 5c). The succinyl-CoA level in ZH40 was hundreds of times lower than that of wild type *E. coli* BW25113, making it too low to determine accurately (Fig. 3d, Supplementary Data 9).

Citrate synthase catalyzes the critical irreversible step that controls the amount of the acetyl-CoA pool and the metabolic flux into TCA cycle. In this study, mutations in citrate synthase reducing the enzyme activity of citrate synthase were found beneficial for the growth recovery of TCA cycle-deficient (ΔaceAΔsucA) *E. coli* (Fig. 1i, Supplementary Table 2, Supplementary Table 8). Tricarboxylic acid metabolic flux was decreased in evolved strain dTCA-E1, comparing to unevolved iTCA strain (Fig. 2a). The mutation in the citrate synthase of strain dTCA-E1 was confirmed able to reduce α-ketoglutarate secretion of TCA cycle-deficient (ΔaceAΔsucA) *E. coli* (Fig. 4f, Supplementary Fig. 5e). A previous work reported that after evolution of the *sdhCB* knock-out *E. coli*, mutations in α-ketoglutarate dehydrogenase were found that reduced tricarboxylic acid metabolic flux and succinate secretion[30]. According to the founding of previous reported work and this study, it was assumed that a glucose-grown TCA cycle-defective *E. coli* strain would tend to minimize carbon waste by reducing the citrate (tricarboxylic acid) metabolism during evolution.

For a heterotrophic microbe like *E. coli*, glucose is entirely oxidized to $CO_2$ and $H_2O$ through the TCA cycle and respiratory chain in aerobic conditions. However, complete oxidation of each acetyl-CoA via the TCA cycle leads to the emission of two $CO_2$, which reduces the carbon flux directed to product formation and negatively affects product yield. Therefore, eliminating the TCA cycle in *E. coli* can decrease carbon dissipation and improve the efficiency of chemicals biosynthesis. To prove this concept, the TCA cycle-deficient DDPYM strain was developed and successfully used to achieve higher yield biosynthesis of chemicals from acetyl-CoA (acetate) and AKG point (G-7-ADCA, glutamate) when compared to strains with an intact TCA cycle (Fig. 4, Supplementary Fig. 5). On the other hand, *E. coli* strains with an intact TCA cycle couldn't prevent carbon loss from acetyl-CoA oxidation, even when their growth is strictly restricted in a whole-cell catalysis solution (Fig. 4e, f, Supplementary Fig. 5a, b). The chassis strain ZH40, with the genotype DDPYM ΔaceAΔsucABCD, accumulated α-ketoglutarate (Fig. 4f, Supplementary Fig. 5c), making it suitable for the efficient biosynthesis of various products that require α-ketoglutarate, such as G-7-ADCA, hydroxy-proline, hydroxy-isoleucine, and glutamate family amino acids. On the other hand, the chassis strain ZH42, with the genotype DDPYM ΔaceAΔsucABCDΔpoxBΔpta gltA(T109P), is a versatile chassis for producing chemicals derived from acetyl-CoA (e.g., butanol and fatty acids) as well as chemicals derived from glycolysis pathway intermediates (e.g., pyruvate). In summary, this work opens up an exciting avenues for more efficient biosynthesis of industrial chemicals in the future.

## Methods

### Plasmid and strain construction
Plasmids information and strains genotypes were listed in Supplementary Table 1. Gene knock-in and knock-out experiments were conducted with reported system[31–37]. Strain dTCA in this work was constructed via deleting genes *gadA* and *gadB* from strain PG05[12]. The

DAOCS gene was cloned from plasmid pDB1S-H7[12] and inserted into plasmid pET28b(+) between NcoI and XhoI sites to form plasmid pET28b(+)-DAOCS (Supplementary Table 1). For overexpression of DAOCS, a section of DE3 cassette (DE3') expressing T7 RNA polymerase was cloned from E. coli BL21(DE3) and integrated into the chromosome of the host strain. Genes gdhA and ppc were cloned from E. coli BW25113 genome and inserted into plasmid pSC2s under Ptac promoter to form plasmid pSC2s-gdhA-ppc (Supplementary Table 1). Genes alsS and alsD were cloned from Bacillus subtilis 168 genome to construct a synthetic operon Ptac-alsSD and insert into the ldhA locus in strain ZH42-alsSD by the CRISPR system[32]. Details for genome and plasmids engineering were listed in Supplementary Data 10.

## Adaptive laboratory evolution

Adaptive laboratory evolution was performed via a series of passages in 50 mL glucose minimal medium (50 mL of M9 minimal medium with 20 mmol/L glucose) in 100 mL flasks shaking at 200 rpm and 37 °C. The TCA cycle-deficient strains pre-cultured overnight in Luria-Bertani medium were transferred to glucose minimal medium. For the first and second transfers, 5 mL culture was transferred into 50 mL minimal medium supplemented with 5 mL and 0.5 mL Luria-Bertani respectively. For the following serial passages, 0.5 mL culture was transferred into 50 mL glucose minimal medium without supplements. Endpoint strains were isolated from the evolutionary population via streaking on glucose minimal medium plate when the growth rate did not increase any more. Clones isolated from the plate were frozen in 150 g/L glycerol and stored at −80 °C for future analysis, such as growth analysis, enzyme activities and genome sequencing. During ALE experiments, GREACE was used to accelerate mutation rate[38], and the relevant mutation plasmid was eliminated during adaptive laboratory evolution without antibiotic selective pressure.

## Mutations analysis

The genome DNA of the evolved strains was extracted and prepared as Illumina PE150 libraries. It was then sequenced using Illumina HiSeq X technology by MajorBio company. The BWA-Samtools-Bcftools SNP calling pipeline[39,40] was employed to identify mutations, with the E. coli BW25113 genome (Accession: CP009273.1) serving as the reference sequence. Only variations that met the criteria of being covered by more than 50 reads and located in the ORF or the upstream 100 bp untranslated region were retained. The genome of unevolved dTCA was also sequenced and analyzed to eliminate any innate genetic variations before ALE experiment. The reads associated with this study have been deposited in the National Center for Biotechnology Information (NCBI) databank SRA, with the BioProject accession number (Accession: PRJNA967749). SNP analysis for the four evolutionary endpoint strains and unevolved dTCA were listed in Supplementary Data 1−6.

Survived clones from two groups of ALE experiments (ALE-1 and ALE-2) were randomly isolated by streaking on glucose M9 minimal medium plates after the second transfer of evolutionary populations in glucose M9 minimal medium supplemented with 0.5 mL Luria-Bertani. For mutation analysis of citrate synthase and succinate dehydrogenase, clones that survived on glucose M9 minimal medium plates were randomly selected for PCR reactions and Sanger sequencing. BW25113 ΔaceAΔsucA, which was pre-cultivated in Luria-Bertani medium, was washed twice with M9 minimal medium and spread on glucose M9 minimal medium plates to isolate surviving clones. The results of the mutation analysis for citrate synthase and succinate dehydrogenase are listed in Supplementary Tables 4−6.

## Growth and metabolite analysis

Unless otherwise noted, growth assays were conducted with 50 mL cultures in 250 mL flasks shaking at 200 rpm and 37 °C. All strains grew in M9 minimal medium with 20 mmol/L glucose and were monitored by measuring the optical density at 600 nm ($OD_{600}$) using a spectrophotometer (Eppendorf BioPhotometer). The $OD_{600}$ values were converted to cell dry weight concentrations using a determined $OD_{600}$-dry cell weight relationship for E. coli (1.0 $OD_{600}$ = 0.32 $g_{DCW}$/L[41]). The initial $OD_{600}$ of the inoculated cultures was 0.01. If a strain was pre-cultivated in Luria-Bertani medium, it should be washed twice with M9 minimal medium before inoculation. Supplements was added as indicated in the text if it's used.

Growth rate was calculated using linear regression of the natural logarithm of $OD_{600}$ versus time. Biomass (or metabolites) yield was calculated using linear regression of biomass dry weight (or metabolites concentration) and glucose concentration in the medium. Glucose uptake rate (mmol/$g_{DCW}$/h) was determined as the ratio of growth rate and biomass yield. The production per gram of dry cell weight (mmol/gDCW) was determined using linear regression on metabolite concentrations and biomass dry weight during log phase of growth.

After centrifugation of the samples, the concentrations of glucose, pyruvate, acetoin, acetate and α-ketoglutarate in the supernatants were determined by high-performance liquid chromatography (HPLC, Agilent 1260 series, Hewlett−Packard), equipped with a 300 mm × 7.8 mm Aminex HPX-87H column (Bio-Rad, Hercules, CA, USA) and a refractive index detector. Analysis was performed at 55 °C with a mobile phase of 5 mmol/L $H_2SO_4$ at a flow rate of 0.6 mL/min. Concentrations of glutamate, G-7-ADCA and penicillin G were measured by HPLC (Agilent 1260 series, Hewlett−Packard), equipped with a ZORBAX Eclipse XDB-C18 column (4.6 mm × 150 mm, 5 μm; Agilen) and a diode-array detector. Glutamate was monitored at 360 nm after DNFB derivation[42], G-7-ADCA and penicillin G were respectively measured at 220 nm and 260 nm[12].

## Determination of intracellular succinyl-CoA level

Strains were cultivated in 20 mmol/L glucose M9 minimal medium to measure the intracellular succinyl-CoA level. Then, at the log phase of growth, the strains were collected by centrifugation at 12,000 × g for 5 min. Subsequently, the collected strains were suspended ($OD_{600}$ = 20) in 1 mL of 80% (vol/vol) methanol (−80 °C) for complete ultrasonication at 4 °C. The resulting supernatant was collected and dried without heat in SpeedVac, while the pellet was redissolved in deionized water. The succinyl-CoA concentration was determined in the redissolved pellet using LC-MS/MS method[35].

## Whole-cell catalysis

Whole-cell catalysis for glucose (50 mmol/L) catabolism was performed in 100 mL flasks at 37 °C and 200 rpm. The cells in the log-phase were collected by centrifugation at 5000 × g for 5 min at 4 °C and suspended in 10 mL Tris-MOPS buffer (100 mmol/L Tris-base and 100 mmol/L MOPS, pH 7.4) at a concentration of about 3.2 $g_{DCW}$/L. The whole-cell catalysis for producing G-7-aminodeacetoxycephalosporanic acid (G-7-ADCA) from penicillin G was conducted with glucose[12]. After 10 h of whole-cell catalysis, the concentrations of acetate, acetoin, pyruvate, and α-ketoglutarate were measured to calculate their molar yields from glucose. The measured molar yields were then divided by their pathway theoretical molar yields and utilized to track carbon flux from glucose to metabolites (fractions of carbon fluxes from glucose to metabolites were calculated and listed in the source data of Fig. 4f). MOPS in the reaction buffer was used as an internal reference for the precise determination of metabolite concentrations, especially when the volume of the whole-cell catalysis was changed.

## Enzyme activity analysis

For enzyme activity analysis, cells in log-phase were collected by centrifugation at 5000 × g for 5 min at 4 °C, suspended in 100 mmol/L Tris-Cl (pH 7.4) buffer, and lysed by sonication. For succinate dehydrogenase assays, 200 μL reaction solution contained 100 mmol/L Tris-Cl (pH 7.4), 2 mmol/L $MgCl_2$, 4 mmol/L sodium succinate, 2 mmol/L

dichlorophenolindophenol, and 10 μL of cell lysate. The linear regression coefficient of $OD_{600}$ value versus time ($R^2 > 0.99$) was used to determine succinate dehydrogenase activity. For citrate synthase assays, 200 μL reaction solution contained 100 mmol/L Tris-Cl (pH 7.4), 2 mmol/L $MgCl_2$, 2 mmol/L oxaloacetic acid, 1 mmol/L acetyl-CoA, 0.65 mmol/L 5,5′-dithiobis-(2-nitrobenzoic acid), and 10 μL of cell lysate. The linear regression coefficient of $OD_{412}$ value versus time ($R^2 > 0.99$) was used to determine citrate synthase activity. The enzyme activity unit (U) was defined as the amount of enzyme required to catalyze the conversion of 1 μmol substrate at room temperature within 1 min. Protein quantification was conducted with coomassie brilliant blue protein assay, for which BSA was used as standard protein sample.

## Chemicals and reagents

All chemicals and media for [13]C-metabolic flux analysis were purchased from Sigma-Aldrich (St. Louis, MO). Isotopic tracers were purchased from Cambridge Isotope Laboratories (Tewksbury, MA): [1,2-[13]C]glucose (99.5%), [1,6-[13]C]glucose (99.5% [13]C), and [4,5,6-[13]C]glucose (99.5% [13]C). The isotopic purity and enrichment of tracers were validated by GC-MS analysis as described[43]. Wolfe's minerals (Cat. No. MD-TMS) and Wolfe's vitamins (Cat. No. MD-VS) were purchased from ATCC (Manassas, VA). The M9 minimal medium used for strains aerobic growth was supplemented with 10 mL/L of Wolfe's minerals and 4.5 mg/L of vitamin B1. Glucose was added as indicated in the text. All media and stock solutions were sterilized by filtration. Commercial reagents for other experiments were purchased from Shanghai Yuanye Bio-Technology Co., Ltd.

## Tracer experiments

For tracer experiments, cells were first precultured overnight at 37 °C in shaker flasks. Cells were then resuspended in fresh medium containing a particular glucose tracer, either [1,2-[13]C]glucose, [1,6-[13]C]glucose, or [4,5,6-[13]C]glucose (10 mmol/L initial concentration). Three parallel labeling experiments were conducted for each strain. The initial $OD_{600}$ of the inoculated cultures was 0.015. After inoculation, cells were grown at 37 °C in parallel mini-bioreactors with a working volume of 10 mL[44]. Air was sparged into the liquid at a rate of 12 mL/min to provide oxygen and to ensure sufficient mixing of the culture by the rising gas bubbles. Cells were collected for GC-MS analysis during the mid-exponential growth phase when the biomass concentration ($OD_{600}$) was between 0.5 and 0.7.

## Gas chromatography-mass spectrometry

GC-MS analysis was performed on an Agilent 7890B GC system equipped with a DB-5MS capillary column (30 m, 0.25 mm i.d., 0.25 μm-phase thickness; Agilent J&W Scientific), connected to an Agilent 5977 A Mass Spectrometer operating under ionization by electron impact (EI) at 70 eV. Helium flow was maintained at 1 mL/min. The source temperature was maintained at 230 °C, the MS quad temperature at 150 °C, the interface temperature at 280 °C, and the inlet temperature at 250 °C. Proteinogenic amino acids was analyzed by GC-MS after *tert*-butyldimethylsilyl (TBDMS) derivation[45]. Labeling of glucose (derived from glycogen) and ribose (derived from RNA) were determined by GC-MS[46,47]. In all cases, mass isotopomer distributions were obtained by integration and corrected for natural isotope abundances[48]. Measurement errors of 0.3% were assumed for all measured mass isotopomers[49].

## Metabolic network model and [13]C-metabolic flux analysis

The metabolic network model used for [13]C-MFA is provided in Supplementary Note 1. The model is based on the *E. coli* model[50], which includes all major metabolic pathways of central carbon metabolism, lumped amino acid biosynthesis reactions, and a lumped biomass formation reaction. Updates to the model include making the reaction between PEP and pyruvate reversible[51], and modeling the dilution of isotopic labeling by atmospheric $CO_2$[52]. [13]C-MFA calculations were performed using Metran software[53], which is based on the elementary metabolite units (EMU) framework[54]. Fluxes were estimated by minimizing the variance-weighted sum of squared residuals (SSR) between the measured and model predicted mass isotopomer distributions and acetate yield using non-linear least-squares regression. All measured mass isotopomers are provided in Supplementary Data 7. For integrated analysis of parallel labeling experiments, the data sets were fitted simultaneously to a single flux model[52,55]. Flux estimation was repeated 10 times starting with random initial values for all fluxes to find a global solution. At convergence, accurate 95% confidence intervals were computed for all estimated fluxes by evaluating the sensitivity of the minimized SSR to flux variations. Precision of estimated fluxes was determined as follows[56]:

$$\text{Flux precision(stdev)} = [(\text{flux}_{\text{upper bound 95\%}}) - (\text{flux}_{\text{lower bound 95\%}})]/4 \quad (1)$$

To describe fractional labeling of metabolites, G-value parameters were included in [13]C-MFA. The G-value represents the fraction of a metabolite pool that is produced during the labeling experiment, while 1-G represents the fraction that is naturally labeled, i.e. from the inoculum[57]. By default, one G-value parameter was included for each measured metabolite in each data set. Reversible reactions were modeled as separate forward and backward fluxes. Net and exchange fluxes were determined as follows:

$$v_{\text{net}} = v_f - v_b \quad (2)$$

$$v_{\text{exch}} = \min(v_f, v_b) \quad (3)$$

To determine the goodness-of-fit, [13]C-MFA fitting results were subjected to a $\chi^2$-statistical test. In short, assuming that the model is correct and data are without gross measurement errors, the minimized SSR is a stochastic variable with a $\chi^2$-distribution[56]. The number of degrees of freedom is equal to the number of fitted measurements $n$ minus the number of estimated independent parameters $p$. The acceptable range of SSR values is between $\chi^2_{\alpha/2}(n-p)$ and $\chi^2_{1-\alpha/2}(n-p)$, where $\alpha$ is a certain chosen threshold value, for example 0.05 for the 95% confidence interval. The transhydrogenase flux and oxidative phosphorylation fluxes in the model are determined using the carbon fluxes estimated from [13]C-MFA[58]. In short, in the metabolic model used for [13]C-MFA, all reactions are redox balanced. The assignment whether NADH, $FADH_2$ or NADPH is oxidized or reduced in any reaction is based on the known biochemistry of *E. coli* enzymes. The total production and consumption fluxes for each cofactor are calculated from [13]C-MFA results and then balanced. To balance NADPH fluxes, a transhydrogenase reaction is included in the metabolic model. To balance NADH and $FADH_2$ fluxes, oxidative phosphorylation reactions are included in the model. The results of flux analysis and cofactor balance analysis were listed in Supplementary Data 8.

## Reporting summary

Further information on research design is available in the Nature Portfolio Reporting Summary linked to this article.

# Data availability

Reads of genome sequencing have been deposited in the National Center for Biotechnology Information (NCBI) databank SRA with the BioProject accession PRJNA967749. Source data are provided with this paper.

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

## Acknowledgements

This work was supported by the National Key R&D Program of China (2022YFC2106000 and 2018YFA0901400, Baixue Lin). C.P.L. and M.R.A. were supported by NSF MCB-1616332 grant.

## Author contributions

H.Z., B.L. and Y.T. conceived and designed the experiments. H.Z., Y.Z., X.X. and Y.X. performed the experiments. C.P. L. and M.R.A. performed the experiments of metabolic flux analysis. All authors contributed to data analysis. B.L., Y.T. and Y. M. supervised the study. H.Z., B.L, M.R.A, and Y.T. wrote the manuscript with contributions from all coauthors.

## Competing interests

The authors declare no competing interests.
