## [Peer Review File · Nature Communications]

A citric acid cycle-deficient *Escherichia coli* as an efficient chassis for aerobic fermentationsReviewers' Comments:

Reviewer #1:

Remarks to the Author:

A citric acid cycle-deficient *Escherichia coli* as an efficient chassis for aerobic fermentations

Zhou et al. have reported manipulation of TCA cycle operation to improve the yield of aerobic fermentation. An *E. coli* strain deficient in the TCA cycle [dTCA] (BW25113 Δ sucA Δ aceA Δ gadA Δ gadB Δ poxB::acs) was not able to grow in a minimal medium; however, upon Adaptive Laboratory Evolution, the strain grew as efficiently as the strain with intact TCA cycle [iTCA]. The evolved strains obtained mutations in TCA cycle enzymes, namely, succinate dehydrogenase and citrate synthase. This rewiring is proposed to be responsible for maintaining the production of succinyl-CoA, which is essential for the biosynthesis of amino acids such as lysine and methionine. On introducing an acetyl-CoA-dependent pathway of lysine/methionine biosynthesis from *Bacillus*, this dependence on succinyl-CoA was relieved, indicating that the succinyl-CoA node was a bottleneck for the growth of the dTCA strain.

Using this finding, they developed an *E. coli* strain (ZH40) having a TCA cycle deficiency and the acetyl-CoA dependent lysine/methionine biosynthetic pathway from *Bacillus*. This strain was modified further for the production of various products such as G-7-ADCA, glutamate, acetate, and acetoin. The study is interesting and might be beneficial for future microbial cell factory-related research. However, the following points should be carefully addressed:

Major Comments

1. Please comment on why genetic blocking of succinate dehydrogenation is a must for succinyl-CoA abundance in the context of an alternative of transcriptional regulation / decreased expression being a possible mechanism for enabling growth. This comment is especially in the context of the complete absence of growth in the dTCA strain.
2. It is not established whether the phenotype of evolved dTCA strains is in response to TCA deficiency or an adaptive evolution response of the parental strain under the same condition. Basically, would you expect to get the same phenotype observed from dTCA if you evolved iTCA similarly? Evolving iTCA and using it for comparison will provide the required resolution. There have been several studies evolving *E. coli* under glucose minimal media (where the TCA cycle is downregulated), but it is unclear the impact of DELTA_poxB and/or the addition copy of acs. Please address as it seems like the variable being tested might not be isolated appropriately using control evolutions.
3. Is the genome sequencing performed on a clone or population? Are there no mutations apart from the listed TCA cycle gene mutations? This information needs to be added and provided, in whole, in a supplementary data set. Furthermore, the methods for sequencing need to be added. How were the mutations identified? Without this information, it is unclear if the authors are correctly linking genotype to phenotype. Are the reads deposited in a public repository for data access?
4. The authors earlier observed a growth inhibitory effect of penicillin G and G-7-ADCA. How is the current set of strains overcoming this toxicity? This should be addressed similarly to the previous study from the same authors.
5. For enzyme activity, normalization is done with regards to the biomass. A heterogeneity in sonication outcome might affect the results. The total protein should be compared among the samples and presented.
6. Figure 2b: How are the CO₂ production levels similar in both the iTCA and dTCA-E1? Should we not expect lesser CO₂ production in the dTCA strain? Did CO₂ levels change in any of the characterized strains? Please elaborate and include any internal flux changes related to CO₂ in the study to support this finding.
7. The work from <https://doi.org/10.3389/fmicb.2018.01793> where ALE was used with a KO of sdhCB in *E. coli* on glucose minimal media is a relevant study which should be compared to the findings in this study as a direct comparison.

Minor Comments

The canonical product of TCA is NADH and not NADPH. Please modify suitably.

[60] Ensure space between E. coli

[128-132] In Figures 1e and 1f in the x-axis legend SdhA and GltA enzyme activity is mentioned; the assay measures the enzyme activity of the entire complex, so please change it to succinate dehydrogenase and citrate synthase enzyme activity.

[138-149] In Figure 1g and 1h, *gltA* should be italicized.

[177] Figure 2c y-axis spelling of 'balance' to be corrected.

Throughout the manuscript, glucose minimal medium and glucose minimum medium have been used interchangeably. Uniform nomenclature would be better.

Figures 2b and 2e are not referred to in the text.

Main figure and corresponding replicates in the supplementary are normalized differently.

Reviewer #2:

Remarks to the Author:

This manuscript deals with the construction of a metabolic chassis, based on a TCA cycle-deficient *E. coli* strain, which could be of interest for the production of various chemicals. The authors first used adaptive laboratory evolution to recover aerobic growth of such a strain in glucose minimal medium and also to identify specific metabolic bottlenecks. Based on sequencing of evolved strain and ¹³C metabolic flux analysis, they found that supply of succinyl-CoA may be the limiting factor of the TCA cycle-deficient *E. coli* strain. This hypothesis was verified and new strains were constructed accordingly, substituting metabolic pathways using succinyl-CoA as precursor, namely for lysine and methionine biosynthesis, by heterologous pathways acetyl-CoA dependent instead. Successfully, although TCA cycle-deficient, these strains recover similar growth rate compared to that of the WT and control strains. The metabolic chassis so-obtained was then tested for the production of 4 chemicals, in which α-ketoglutarate is the precursor for both of them.

Overall, the results are technically sound and the data of high quality. The level of support of authors' conclusions is strong and the significance of the results is of interest in the field. From my point of view, the novelty of this work is related to the metabolic engineering strategy to circumvent the supply in succinyl-CoA. Indeed, *sdhA* mutations in the evolved strains may be anticipated in the light of previous works decades ago reporting revertant of *sucA* and *lpd* mutants in which the succinate dehydrogenase is inactivated (Creaghan & Guest, 1977 and 1978). Mutation in *gltA* leading to an attenuated form of the citrate synthase is much more unexpected. However, its significance remains elusive after this study. The authors did not analyze it deeply further because it did not appear as essential for the adaptation of *E. coli* to a defect of its TCA cycle.

Major concerns:

1/ Growth of the Δ *sucA* Δ *aceA* strains.

In the introduction, the authors state that "inactivation of inactivation of α-ketoglutarate dehydrogenase (*ΔsucA*) and glyoxylate shunt (*ΔaceA*) 1-3, 5 resulted in no aerobic growth on glucose." (L44). However, from my reading of cited articles, it is not clearly reported that the Δ *sucA* Δ *aceA* strain does not grow in that condition.

Furthermore, adaptive laboratory evolution (ALE) was performed on a *BW25113ΔsucAΔaceAΔgadAΔgadBΔpoxB::acs* "to restore aerobic growth in glucose minimal medium" (L84). That strain is supposed to be not able to grow in such condition. How it is feasible to perform such an experiment if the strain is not growing?

This should be clarified all along the manuscript.

2/ Succinyl-CoA supply.

Authors hypothesized that the "the causal bottleneck for growth of TCA cycle-deficient *E. coli* was the

biosynthesis of lysine and methionine due to insufficient succinyl-CoA supply". They circumvented successfully this issue by adding in the medium lysine and methionine, or constructing strains that do not require succinyl-CoA for the biosynthesis of lysine and methionine. However, this remains indirect evidence for succinyl-CoA supply. Quantification of succinyl-CoA pool in the WT strain and in growing deficient-TCA cycle strains should provide the evidence that the latter strains are able to grow despite of a low (or inexistent) level of succinyl-CoA in the cells.

3/ Use of Δ sucABCD strain

"2.5 TCA cycle-deficient E. coli as an efficient chassis strain for aerobic fermentations"

The authors decided to move toward the use of a Δ aceA Δ sucABCD chassis instead of pursuing the work with Δ aceA Δ sucA chassis. In the former chassis, two (consecutive) TCA cycle enzymes are inactivated instead of one. There is no clear explanation for this chassis change and no data with this strain in this study is provided earlier in the manuscript. Particularly, if we guess that succinyl-CoA is also limiting for lysine and methionine biosynthesis in such a strain, the impact of sucCD deletion in addition to that of sucA was not investigated. There are potential other effects of this double deletion on metabolism.

Minor concerns:

L21 - "improved biosynthesis yields." Of what?

L22 - "universal chassis for biosynthesis." Of what?

L39 - "the oxidation of acetyl-CoA to CO₂ generates (ATP) and reductants (NADPH) ...": This generates mainly reductants as NADPH (1 mole) and especially FADH₂ (1 mole) and NADH (2 moles). The coupling with oxidative phosphorylation in aerobic condition provide indeed much more ATP moles than the unique mole of ATP generated in the TCA cycle.

L135 - "was replaced with the identified mutants." Mutant genes?

L147 - "iTCA strain, which has a complete TCA cycle". iTCA is BW25113 Δ poxB::acs. Why using that strain and not the WT strain?

L192 - "lead to ATP shortages". Reference?

L213 - meso-2,6-diaminoheptanedioate: Does the strain grow without adding that compound in the mixture? If no, why? If yes, why supplementing the medium with it?

L255 - "However, that strain could not grow in glucose minimum medium." Not clearly mention in the cited reference. It looks like the production of deacetoxycephalosporin C is indeed toxic.

L298 - What is the benefit of using a TCA cycle-deficient strain? There is no comparison with counterpart strains having a complete TCA cycle.

L328 - "It has long been believed that a complete TCA cycle is necessary for aerobic respiration and energy production in heterotrophic microbe." Adding a reference would be better.

Figure 1: The specific glucose consumption rates would be also of interest.

Figure 2: The choice of colors may complicate the analyze of the figure for colorblind readers (same for supplementary figures).

Reviewer #3:

Remarks to the Author:

Zhou et al constructed an E. coli strain in which the TCA cycle is not functioning, using a combination of knockouts and lab evolution. They propose that the resulting dTCA strain is an improved production host for biotechnological processes. They saw that succinyl-CoA is the bottleneck for growth of a TCA deficient strain and by replacing succinyl-CoA utilizing enzymes with AcCoA dependent enzymes another strain (the DDPYM strain) was improved. Overall this is an interesting manuscript, it is clearly writing and addresses an important question: how can we improve metabolism of industrial microbes. However, I have some major points that should be addressed, especially the motivation of deleting the TCA cycle and the mechanistic insights of why the dTCA strain is better.

1) Rational for deleting the TCA cycle. The authors say in the introduction that "it is believed that blocking the TCA cycle and its bypass pathways could reduce carbon losses and improve product yields in aerobic fermentations." This is not really clear, because the TCA cycle is much more efficient in generating energy than fermentation. The authors should clearly explain why deletion of the TCA cycle has benefits. I understand that the yield improves in non- or slow-growing strains but a growth arrest can be achieved by deletion of many enzymes and other cellular processes. What is the exact mechanism that makes the dTCA strain a better host?

2) Evolution experiment. The starting strain does not grow (in Figure 1a dTCA has a growth rate of 0), but the authors say that it needs 230 generations for evolution. How does a non-growing strain divide 230 times?

3) Whole genome sequencing (WGS). The authors performed WGS to identify the ALE mutations. However the results are not clearly presented and only mutations in *sdhA* and *gltA* are presented. Were these the only mutations? The authors should attempt a more systematic analysis to understand if there are adaptive mutations in other pathways.

4) Interpretation of the Flux Analysis. In the flux map of the dTCA strain it is unclear which reaction produces succinate. The authors should also explain if all of the fluxes are balanced and how they included biomass production in the model.

5) Claim that TCA cycle is not vital for energy generation is misleading. The authors report that ATP formation is higher in the dTCA strain, but it is not clear why this supports the claim that the TCA cycle is not vital for energy production. It only shows that this strain has a higher demand for ATP. What is the reason for the higher ATP demand? Are there futile cycles that waste ATP?

6) Replacing SucCoA dependent pathways. The motivation for replacing the SucCoA enzymes is not clear. First the authors evolved the dTCA strain and found that deleting *sdhA* in the *sucA* background improves the strain. Do I understand it correctly that now the goal is to use a second approach to improve the dTCA strain by rational design that resulted in the DDPYM strain? This part should be better connected with the evolution part, especially because the DDPYM strain is used as production host (so why doing the evolution in the first place).

7) Overproduction of chemicals. The authors show mostly titers (g/L), but to show that flux into the product increases they should report fluxes in g/L per g Biomass. Overall the section is difficult to understand because different genetic modifications are made for the different products and it is unclear which modification results in the improved production rates. I suggest that the authors highlight better the advantage of the TCA deletion, e.g. higher substrate levels at the AKG branch point.

8) The authors don't provide insights about the mechanisms that improved production in the TCA deficient strain. This is reflected in a very brief discussion, at least here the authors could provide hypothesis about why a TCA deletion is beneficial for overproduction.

The manuscript “ A citric acid cycle-deficient *Escherichia coli* as an efficient chassis for aerobic fermentations ” (NCOMMS-23-15866A) has been carefully revised. We appreciate the kind instructions from the editor and the invaluable and detailed comments and suggestions from the reviewers. The point-by-point responses to the reviewers’ comments are listed below.

List of actions (main changes made in the manuscript)

LOA1: Glucose uptake rate was added in as Figure 1d (line 122) and relevant sentence (line 120). Therefore, Figure 1d-h in previous manuscript was renamed as Figure 1e-i accordingly.

LOA2: The title of *x*-axis in Figure 1f-g was changed to succinate dehydrogenase and citrate synthase enzyme activity in lysate. And the enzyme activity had been normalized as U/mg total protein. The description of the enzyme activity was also revised to U/mg total protein in the relevant text. (line 150)

LOA3: The *gltA* (in previous Figure 1g-h) has been italicized in revised Figure 1h-i.

LOA4: The colors of Figure 2 and Supplementary Figure 2 were modified to be friendly to color blind readers. Figure 2b and Supplementary Figure 2a were modified to show more detailed information on the internal flux changes related to CO₂, relevant sentences were added in text (line 186).

LOA5: The spelling ‘balance’ in the title of *y*-axis in Figure 2c (line 189) has been corrected.

LOA6: Figure 3d (line 285) and relevant sentences (line 253, line 275-279) were added in “Results” to show the intracellular succinyl-CoA level in wild type and engineered *E. coli* strains.

LOA7: The previous Figure 4f (Production of acetoin during aerobic growth in glucose minimal medium) was deleted in revised manuscript. The previous Figure 4g was renamed as Figure 4f in the revised manuscript, and its figure legend was changed to “Fraction of carbon flux from glucose to metabolites during whole-cell catalysis”. Fraction of metabolites carbon flux in strain ZH40 was added in revised Figure 4f.

LOA8: A new Supplementary Figure 1 and relevant sentences (line 180, line 398) were added to show the potential replenishment pathway for succinate through L-aspartate oxidase (NadB) and fumarate reductase (FRD) reactions. Figure S1-Figure S3 in previous manuscript were renamed as Supplementary Figure 2 – Supplementary Figure 4 accordingly, of which the previous Figure S3c (aerobic growth of acetoin producing strains) was deleted in revised Supplementary Figure 4.

LOA9: Supplementary Figure 5 was supplemented to show the process of whole-cell catalysis from glucose to metabolites.

LOA10: Supplementary Data 1 was supplemented to show the aerobic growth of evolved and engineered dTCA strains in glucose minimal medium.

LOA11: In Supplementary Table 1 (line 43 in Supplementary Materials), those strains (ZH41, BW25113-alsD, DDPYM-alsD, ZH40-alsD, ZH41-alsD, ZH42-alsD) and plasmids

(pSC2s-alsD, pSC2s-alsSD) which were not mentioned in revised manuscript were deleted.

LOA12: Data about the production of acetoin in strains (BW25113-alsD, DDPYM-alsD, ZH40-alsD, ZH41-alsD, ZH42-alsD) were deleted in revised Supplementary Table 5 (line 63 in Supplementary Materials). Sentences about the results of acetoin production were also removed in the revised manuscript.

LOA13 : The genomic mutations in evolved dTCA strains were supplemented as Supplementary Data 2. Reads of genome sequencing was deposited in a public repository for data access (BioProject accession: PRJNA967749) (line 137, line 495).

LOA14: “GCMS Data” was moved from previous Supplementary Data 2 to Supplementary Data 4 in the revised manuscript. “Flux analysis results” was moved from previous Supplementary Data 1 to Supplementary Data 5 in the revised manuscript.

LOA15: Supplementary Data 6 was added to list mutations of citrate synthase and succinate dehydrogenase in survived dTCA and BW25113 *ΔaceAΔsucA* clones. (line 234, line 404)

LOA16: Intracellular succinyl-CoA level was supplemented in Supplementary Data 7. (line 254, line 275- 279)

LOA17: The previous Table S6 was replaced with Supplementary Data 8, and sentences about results of carbon flux tracking from glucose to metabolites were revised. (line 339-368)

LOA18: “glucose minimum medium” was changed to “glucose minimal medium” throughout the revised manuscript. “*ΔsucAΔaceA*” was changed to “*ΔaceAΔsucA*” throughout the revised manuscript.

LOA19: Spelling of *E. coli* was checked and correct throughout the revised manuscript.

LOA20: Units “mM, $\text{g}_{\text{DCW}} \cdot \text{g}_{\text{glc}}^{-1}$, $\text{mmol} \cdot \text{g}_{\text{DCW}}^{-1} \cdot \text{h}^{-1}$, $\text{mmol} \cdot \text{g}_{\text{DCW}}^{-1}$ and $\text{mol} \cdot \text{mol}^{-1}$ ” was changed to “mmol/L, $\text{g}_{\text{DCW}}/\text{g}_{\text{glc}}$, $\text{mmol}/\text{g}_{\text{DCW}}/\text{h}$, $\text{mmol}/\text{g}_{\text{DCW}}$ and mol/mol ” throughout the revised manuscript.

LOA21: We had supplemented sentences to explain that succinate dehydrogenase mutation is necessary and inevitable, and highlight the advantage of TCA cycle deficiency (*ΔaceAΔsucA*) on chemicals production in the section of “Discussion”. (line 389-455)

LOA22: Sentences “Heterotrophic bacteria rely on the tricarboxylic acid cycle (TCA cycle) for aerobic growth. Theoretically, eliminating the TCA cycle would decrease carbon dissipation and facilitate improved biosynthesis yields.” (line 20 in previous manuscript) were revised to “Tricarboxylic acid cycle (TCA cycle) plays an important role for aerobic growth of heterotrophic bacteria. Theoretically, eliminating TCA cycle would decrease carbon dissipation and facilitate chemicals biosynthesis.” (line 21)

LOA23: We had modified sentences to explain why deletion of the TCA cycle would be beneficial to biosynthesis. (line 41-47)

LOA24: Although penicillin G and G-7-ADCA was toxic, they were not included in the growth medium. Details of the whole-cell biocatalytic process for G-7-ADCA production were added according to reviewer’s suggestion (line 73-83).

LOA25: In adaptative laboratory evolution (ALE), the strain dTCA was pre-cultivated in Luria-Bertani medium and subsequently transferred to glucose minimal medium for passages.

We added sentences to describe more details of the ALE process (line 95-99). The detailed method of ALE assay was revised in the “Methods” (line 472).

LOA26: We added sentences to explain why the pyruvate oxidase gene *poxB* was replaced with the acetyl-CoA synthetase gene *acs* (Δ *poxB::acs*) (line 104).

LOA27: The statement about function of TCA cycle in energy generation in strain dTCA-E1 was revised in the revised manuscript (line 213, line 219)

LOA28: Parts of “Results” (line 313-368) and discussion (line 389-455) were modified to highlight the advantage of TCA cycle deficiency to chemicals production.

LOA29: In “Methods”, “Mutations analysis” (line 485-505), “Determination of intracellular succinyl-CoA level” (line 529-536) and “Whole-cell catalysis” (line 537-550) were added. Writing of “Methods” was polished to show more details of assays in this work. Supplementary Data 9 was added to show the details of genome and plasmids engineering. (line 470)

LOA30: Citations of figures, tables and references in text were carefully checked and cited in revised manuscript. References were updated.

LOA31: The writing of the manuscript was polished. Grammatical errors had been revised.

LOA32: A coauthor was added in the author list (line 4), who carried out the experiments (Quantification of succinyl-CoA pool) that the reviewer suggested. The other authors have no competing interests to declare.

Responses to the reviewers' comments

To Reviewer #1:

Comment 1: Please comment on why genetic blocking of succinate dehydrogenation is a must for succinyl-CoA abundance in the context of an alternative of transcriptional regulation / decreased expression being a possible mechanism for enabling growth. This comment is especially in the context of the complete absence of growth in the dTCA strain.

Response 1:

Thank you for your kind comment.

We acknowledge that transcriptional regulation of succinate dehydrogenase might be alternative mechanisms. In this study, the genome sequence analysis of the evolved dTCA strains showed that genes encoding for succinate dehydrogenase were mutated in all the evolutionary endpoint strains (Figure 1e and Supplementary Data 2). The enzymatic activity analysis revealed that succinate dehydrogenase mutants had no detectable enzymatic activity (Figure 1f). Additionally, the genome sequence analysis showed that there was no mutation in the upstream 5'UTR which might affect transcription of succinate dehydrogenase operon *sdhCDAB* (see: <https://www.ncbi.nlm.nih.gov/sra/PRJNA967749>, and Supplementary Data 2).

In the dTCA strain, the TCA cycle was blocked at α -ketoglutarate dehydrogenase (*AsucA*), resulting in the exclusive regeneration of succinyl-CoA from succinate. We hypothesized that, once the glyoxylate cycle was simultaneously deleted at the isocitrate lyase node (*lAceA*), the availability of succinyl-CoA in dTCA would be inversely correlated with the activity of succinate dehydrogenase. This would create a strong selective pressure for the mutation of succinate dehydrogenase during adaptive evolution.

We also sequenced *sdhCDAB* of the survived TCA cycle-deficient strains (dTCA and BW25113 $\Delta aceA \Delta sucA$) clones on glucose minimal medium plate. Results showed that all the randomly isolated 30 clones contained at least one mutation at the ORF of *sdhCDAB* (Supplementary Data 6). This suggests that succinate dehydrogenase mutation was inevitable for relieving the growth restriction of TCA cycle-deficient *E. coli*.

Table 1. Mutation analysis of citrate synthase and succinate dehydrogenase in dTCA clones from ALE-1 culture that survived on glucose M9 medium plate. (Supplementary Data 6 Sheet1)

Clones	GltA	SdhC	SdhD	SdhA	SdhB
1				E 62 D	
2				L 369 F	
3			F 51 FS		
4	H 130 Y			D 6 Y	
5				T 522 M	
6				E 305 K	
7					Q 126 K
8				A 34 T, S 536 F	
9				E 62 D, G 416 D, L 42 P	
10				G 51 S	

FS: Frame shift.

Table 2. Mutation analysis of citrate synthase and succinate dehydrogenase in dTCA clones from ALE-2 culture that survived on glucose M9 medium plate. (Supplementary Data 6 Sheet2)

Clones	GltA	SdhC	SdhD	SdhA	SdhB
1				L 408 P	
2				G 51 S	
3				L 408 P	
4				P 222 FS	
5				P 222 FS	
6				C 257 Y	
7				T 134 I	
8				G 51 S	
9				D 287 FS	
10				G 51 S	

FS: Frame shift.

Table3. Mutation analysis of citrate synthase and succinate dehydrogenase in BW25113 $\Delta aceA\Delta sucA$ clones that survived on glucose M9 medium plate. (Supplementary Data 6 Sheet3)

Clones	GltA	SdhC	SdhD	SdhA	SdhB
1					L 73 FS
2				A 135 FS	
3			sdhD::insB1		
4				S 529 Y	
5		K 7 FS			
6					L 73 FS
7					Q 135 Stop
8					L 73 FS
9				E 237 Stop	
10				G 103 D	

FS: Frame shift.

Stop: Stop codon.

insB1: insertion sequence *insB1*.

According to the reviewer's suggestion, Supplementary Data 6 was supplemented to show the result of mutation analysis of the survived TCA cycle-deficient strains (dTCA and BW25113 $\Delta aceA\Delta sucA$) on glucose minimal medium plate. The relevant sentences were added in the revised manuscript: "Succinate dehydrogenase mutations were found in all 30 randomly isolated clones (mutations analysis of citrate synthase and succinate dehydrogenase were listed in Supplementary Data 6) when we sequenced the succinate dehydrogenase operon in dTCA or BW25113 $\Delta aceA\Delta sucA$ clones that survived on glucose M9 minimal medium plate. This finding indicated that succinate dehydrogenase mutation was seemingly inevitable." (line 235).

Additionally, we supplemented some sentences in the revised manuscript to discuss why the succinate dehydrogenase mutation was necessary and inevitable for the aerobic growth of TCA cycle defected ($\Delta aceA\Delta sucA$) *E. coli* in "Discussion", as follow: "Adaptive laboratory evolution was used to gain a better understanding of the underlying mechanisms causing the

growth defect observed when TCA cycle was severed at the α -ketoglutarate dehydrogenase step in *E. coli*. It was discovered that the limiting factor for growth was not energy limitation but insufficient supply of succinyl-CoA resulting in the inability to produce three amino acids, methionine, DAP and lysine. Succinyl-CoA can only be generated from succinate in *E. coli* if the α -ketoglutarate dehydrogenase reaction was blocked (*AsucA*). When the glyoxylate cycle was also blocked by inactivation of isocitrate lyase (*AceA*), the *E. coli* with defective TCA-cycle (*DeltaAceAAsucA*) lacked sufficient succinate to maintain the succinyl-CoA pool. Although succinate might be supplemented from fumarate by L-aspartate oxidase (NadB) or fumarate reductase (FRD) (Supplementary Figure 1), the amount of succinate was limited. Due to succinate being rapidly oxidized by succinate dehydrogenase, TCA cycle defected (*DeltaAceAAsucA*) *E. coli* could not maintain a sufficient succinyl-CoA pool without enough intracellular succinate, unless its succinate dehydrogenase was mutated. Upon transfer to glucose minimal medium plates from rich medium, those survived dTCA and BW25113 *DeltaAceAAsucA* clones were found to have succinate dehydrogenase mutations (Supplementary Data 6). This work provides evidence that succinate dehydrogenase mutation is necessary and unavoidable for recovering aerobic growth of TCA-cycle defective *E. coli* (*DeltaAceAAsucA*) in glucose minimal medium.” (line 389-406)

*Comment 2: It is not established whether the phenotype of evolved dTCA strains is in response to TCA deficiency or an adaptive evolution response of the parental strain under the same condition. Basically, would you expect to get the same phenotype observed from dTCA if you evolved iTCA similarly? Evolving iTCA and using it for comparison will provide the required resolution. There have been several studies evolving *E. coli* under glucose minimal media (where the TCA cycle is downregulated), but it is unclear the impact of DELTA_poxB and/or the addition copy of acs. Please address as it seems like the variable being tested might not be isolated appropriately using control evolutions.*

Response2:

Thank you for your kind comment.

We acknowledge that if we evolve iTCA in the beginning and use it for comparison, it would enable us to determine whether the phenotype of evolved dTCA strains is a response to TCA deficiency or an adaptive evolution response of the parental strain under the same condition. The comparative genomic analysis and MFA analysis of the evolved dTCA strains with iTCA only elucidated the mechanism of growth recovery in TCA cycle-deficient *E. coli*, but not the *E. coli*'s adaptation to TCA deficiency. According to the reviewer's suggestion, we had deleted the phrases "adapted to TCA cycle defect" and "the adaptation of *E. coli* to a TCA cycle defect" in the revised manuscript and made appropriate revisions to the related sentences.

In this work, we aimed to identify the cause for growth arrest in TCA cycle-deficient *E. coli* (dTCA strain) and elucidate the mechanism by which the dTCA strain can **restore its aerobic growth to the level of an unevolved *E. coli* with an intact TCA cycle**. In previous work, we successfully increased the yield of G-7-aminodeacetoxycephalosporanic acid (G-7-ADCA) from penicillin G by blocking the TCA cycle in *E. coli* (Δ *sucA* Δ *aceA*) and reconstituting it with α -ketoglutarate-dependent deacetoxycephalosporin C synthase (DAOCS) (Proc Natl Acad Sci USA (2015) 112, 9855-9859). This approach of reconstituting the TCA cycle with enzymes that catalyze TCA cycle-coupling reactions could be utilized for other enzymes as well. However, we encountered a challenge in that study as we could not cultivate the TCA cycle-deficient strain in glucose minimal medium. Therefore, we conducted adaptive laboratory evolution on *E. coli* strain dTCA to determine if its aerobic growth in glucose minimal medium could be restored. We did not subject the control strain iTCA to the same evolutionary process.

To estimate if the TCA cycle in the dTCA strain remained blocked during evolution, we replaced the pyruvate oxidase gene *poxB* with the acetyl-CoA synthetase gene *acs* (Δ *poxB*::*acs*). This replacement ensured that acetate could only generate from acetyl-CoA and could be readily converted back to the acetyl-CoA pool. Consequently, we monitored the acetate (acetyl-CoA) pool to determine if the TCA cycle was still blocked in the dTCA. Therefore, we used iTCA (BW25113 Δ *poxB*::*acs*) instead of the wild type BW25113 as a control to perform MFA analysis in order to investigate the metabolic mechanisms involved in the growth recovery of evolved dTCA (BW25113 Δ *sucA* Δ *aceA* Δ *gadA* Δ *gadB* Δ *poxB*::*acs*).

According to the reviewer's suggestion, we had carefully revised the related sentences which were listed as below:

(1) Figure legend of Figure 1 "*Escherichia coli* adapted to TCA cycle defect with succinate dehydrogenase mutation and recovered aerobic growth in glucose minimal medium" was revised to "The TCA cycle-defective *Escherichia coli* strain (dTCA) showed aerobic growth recovery in glucose minimal medium when a succinate dehydrogenase mutation was introduced." (line 123)

(2) "These results suggest that inactivation of SdhA and attenuation of GltA were critical for *E. coli* adaptation to a TCA cycle defect." was revised to "These results suggested that inactivation of succinate dehydrogenase and attenuation of citrate synthase were critical for growth recovery of TCA cycle defected *E. coli*." (line 152)

(3) "These results suggest that SdhA inactivation was essential for the adaptation of *E. coli* to a TCA cycle defect, while GltA attenuation although not essential was beneficial." was revised to "These results suggested that succinate dehydrogenase inactivation was essential for the growth recovery of TCA cycle defected *E. coli*, while citrate synthase attenuation although not essential was beneficial." (line 162)

(4) “To elucidate the metabolic mechanisms of *E. coli* adaptation to a disrupted TCA cycle, ^{13}C -metabolism flux analysis (^{13}C -MFA) was performed on the evolved dTCA-E1 strain and iTCA strain, which has a complete TCA cycle.” was revised to “To elucidate the metabolic mechanisms for growth recovery of evolved TCA cycle-deficient *E. coli*, ^{13}C -metabolism flux analysis (^{13}C -MFA) was performed on the evolved dTCA-E1 strain and iTCA strain, which had a complete TCA cycle (Date of MFA were shown in Supplementary Data 5).” (line 168)

(5) “The pyruvate oxidase gene *poxB* was replaced with acetyl-CoA synthetase gene *acs* (Δ *poxB*::*acs*) in this strain so that acetate could be more efficiently converted back to acetyl-CoA.” was revised to “In order to monitor the acetate (acetyl-CoA) pool and determine if the TCA cycle in the dTCA strain remained blocked during evolution, the pyruvate oxidase gene *poxB* was replaced with the acetyl-CoA synthetase gene *acs* (Δ *poxB*::*acs*). This replacement ensured that acetate could only be formed through acetyl-CoA and efficiently converted back to the acetyl-CoA pool.”(line 104)

Comment 3: *Is the genome sequencing performed on a clone or population? Are there no mutations apart from the listed TCA cycle gene mutations? This information needs to be added and provided, in whole, in a supplementary data set. Furthermore, the methods for sequencing need to be added. How were the mutations identified? Without this information, it is unclear if the authors are correctly linking genotype to phenotype. Are the reads deposited in a public repository for data access?*

Response 3:

Thank you for your kind comment.

Genome sequencing of strain dTCA, dTCA-E1, dTCA-E2, dTCA-E3, dTCA-E4 was performed on a clone. We have added detail in the “Methods” part: “Endpoint strains were isolated from the evolutionary population via streaking on glucose minimal medium plate when the growth rate did not increase any more. Clones isolated from the plate were frozen in 150 g/L glycerol and stored at -80°C for future analysis, such as growth analysis, enzyme activities and genome sequencing.” (line 480).

In addition to the mentioned TCA cycle gene mutations, numerous other mutations were identified. Details of all these mutations can be found in Supplementary Data 2. Sentence “Several genes coding for enzymes in the TCA cycle were found to be mutated in the evolved strains” was revised to “The analysis of gene mutations in the evolved strains revealed common mutations in genes encoding enzymes in the TCA cycle (Figure 1e, Supplementary Data 2).” (line 141)

According to the reviewer’s suggestion, we added the detail of genome sequencing and mutations identification in the method: “The genome DNA of the evolved strains was

extracted and prepared as Illumina PE150 libraries. It was then sequenced using Illumina HiSeq X technology by MajorBio company. The BWA-Samtools-Bcftools SNP calling pipeline was employed to identify mutations, with the *E. coli* BW25113 genome (Accession: CP009273.1) serving as the reference sequence. Only variations that met the criteria of being covered by more than 50 reads and located in the ORF or the upstream 100 bp untranslated region were retained. The genome of unevolved dTCA was also sequenced and analyzed to eliminate any innate genetic variations before ALE experiment. The reads associated with this study have been deposited in the National Center for Biotechnology Information (NCBI) databank SRA, with the BioProject accession number (Accession: PRJNA967749). SNP analysis for the four evolutionary endpoint strains were listed in Supplementary Data 2. ”
(line 486-496)

Comment 4: *The authors earlier observed a growth inhibitory effect of penicillin G and G-7-ADCA. How is the current set of strains overcoming this toxicity? This should be addressed similarly to the previous study from the same authors.*

Response 4:

Thank you for your kind comment.

A whole-cell biocatalytic process was developed for converting penicillin G to G-7-ADCA, based on an engineered *Escherichia coli* strain expressing α -ketoglutarate-dependent deacetoxycephalosporin C synthase (DAOCS). The biotransformation process consisted of two stages: cell growth and substrate conversion. In the cell growth stage, a TCA deficient strain was cultured and induced for overexpression of deacetoxycephalosporin C synthase, without the presence of penicillin G. This was followed by the conversion stage, where resting cells were used as biocatalyst and penicillin G served as the substrate. Both penicillin G and deacetoxycephalosporin C (G-7-ADCA) were found to inhibit cell growth of *E. coli*. In the cell growth stage, there was no penicillin G or G-7-ADCA in the medium. During the conversion stage, the resting cells were used, and there was no cell growth.

According to the reviewer's suggestion, we had revised relevant sentences.

Sentences “In previous work, we blocked the TCA cycle in *E. coli* (Δ *sucA* Δ *aceA*) and reconstituted it with α -ketoglutarate-dependent deacetoxycephalosporin C synthase (DAOCS), successfully increasing the yield of G-7-aminodeacetoxycephalosporanic acid (G-7-ADCA) from penicillin G. However, since both the substrate and products of DAOCS were antibiotics and toxic to *E. coli*, we could not cultivate this engineered strain in glucose minimal medium. However, since both the substrate and products of DAOCS were antibiotics and toxic to *E. coli*, we could not cultivate this engineered strain in glucose minimal medium.” were revised as “In previous work, a whole-cell biocatalytic process was developed that employed an engineered TCA cycle-deficient (Δ *sucA* Δ *aceA*) *E. coli* strain expressing α -ketoglutarate-

dependent deacetoxycephalosporin C synthase (DAOCS) to convert penicillin G to G-7-ADCA. Initially, the TCA cycle-deficient strain was cultured with a rich medium and induced for overexpression of deacetoxycephalosporin C synthase, without the presence of penicillin G, during the cell growth stage. Subsequently, the resting cells, which served as the biocatalyst, were used to produce deacetoxycephalosporin C with penicillin G functioning as the substrate¹². However, due to the toxic nature of both the substrate and products of DAOCS on *E. coli*, it was impossible to repair the TCA cycle using the DAOCS reaction while cultivating this engineered strain in a glucose minimal medium.”(line 73-82).

Sentence “However, that strain could not grow in glucose minimal medium.” was revised to “As penicillin G and G-7-ADCA was toxic to *E. coli*, the defective TCA cycle could not be repaired with α -ketoglutarate-dependent DAOCS to support cell growth in glucose minimal medium.” (line 303)

Comment 5: For enzyme activity, normalization is done with regards to the biomass. A heterogeneity in sonication outcome might affect the results. The total protein should be compared among the samples and presented.

Response 5:

Thank you for your kind comment.

According to the suggestion, the enzyme activity had been normalized as U/mg total protein (Figure 1f-g). The description of the enzyme activity was also revised to U/mg total protein in the relevant text.

Sentences “The SdhA mutants R133H and G51S had no detectable succinate dehydrogenase activity (Figure 1e). The activities of GltA(T109P) (0.035±0.003 U/g_{DCW}), GltA(L10I, T109P) (0.035±0.001 U/g_{DCW}), GltA(H157Y) (0.073±0.002 U/g_{DCW}) and GltA(I375T) (0.035±0.001 U/g_{DCW}) were significantly attenuated compared to the control strain iTCA (0.098±0.005 U/g_{DCW}) (Figure 1f)” were corrected as “No significant succinate dehydrogenase activity was detected in the lysate of the four evolved strains containing succinate dehydrogenase mutations (Figure 1f). The activities of GltA(T109P) (0.019±0.002 U/mg total protein), GltA(L10I, T109P) (0.019±0.001 U/mg total protein), GltA(H157Y) (0.040±0.001 U/mg total protein) and GltA(I375T) (0.019±0.001 U/mg total protein) were significantly attenuated compared to the control strain iTCA (0.054±0.003 U/mg total protein) (Figure 1g).”(line 149).

Method for protein quantification was added in the “Methods” part: “Protein quantification was conducted with coomassie brilliant blue protein assay, for which BSA was used as standard protein sample.” (line 564).

Comment 6: Figure 2b: How are the CO₂ production levels similar in both the iTCA and

dTCA-E1? Should we not expect lesser CO₂ production in the dTCA strain? Did CO₂ levels change in any of the characterized strains? Please elaborate and include any internal flux changes related to CO₂ in the study to support this finding.

Response 6:

Thank you for your kind comment.

The carbon balance (C mol/C mol) was calculated to determine the amount of carbon flux to CO₂. The CO₂ production levels from the TCA cycle, acetate formation, and other metabolic processes were analyzed in Supplementary Data 5 and Figure 2b. The results indicated that although the total CO₂ production levels were similar in the iTCA (27.3%) and dTCA-E1 (27.4%) strains, the CO₂ production from the TCA cycle was significantly lower in the dTCA-E1 strain (2.2%) compared to the iTCA strain (11.7%). Furthermore, in the dTCA-E1 strain, acetate formation from pyruvate resulted in higher CO₂ production (17.9%) than in the iTCA strain (10%). This observation suggests that the CO₂ production from the TCA cycle is significantly lower in the dTCA-E1 strain than in the iTCA strain.

According to the reviewer's suggestion, we had provided more detailed information on the internal flux changes related to CO₂ in the revised version of Figure 2b, including the CO₂ production levels from the TCA cycle, acetate formation, and other metabolic processes. (line186)

Figure 2b Carbon balance analysis. Co-factor metabolism balance analysis was shown as the normalized production (positive value) and consumption (negative value) per 100 mol of glucose consumed. **dTCA-E1**: evolved TCA cycle deficient dTCA strain; **iTCA**: strain with an intact TCA cycle (BW25113 *ApoxB::acs*). Data from three independent biological repeat experiments were used for analysis.

We had added sentences to explain internal flux changes to CO₂: “In the dTCA-E1 strain, only 2.2% of carbon was converted into CO₂ via the TCA cycle, whereas in the iTCA strain, this percentage was much higher at 11.7% (Figure 2b, Supplementary Data 5). Notably, CO₂ production in the dTCA-E1 strain was primarily associated with the synthesis of α -ketoglutarate. Most of the CO₂ released by dTCA-E1 was generated during the formation of acetate from pyruvate (Supplementary Data 5).” (line 186)

Table 4. Carbon balance analysis (C mol/C mol)

	iTCA strain	dTCA-E1 strain
Biomass production	52.7%	36.8%
Acetate production	20.0%	35.9%
Total CO ₂ production	27.3%	27.4%

CO ₂ from TCA cycle	11.7%	2.2%
CO ₂ from Acetate formation	10.0%	17.9%
CO ₂ from other metabolism	5.6%	7.2%

Data from Supplementary Data 5.”

Comment 7: *The work from <https://doi.org/10.3389/fmicb.2018.01793> where ALE was used with a KO of *sdhCB* in *E. coli* on glucose minimal media is a relevant study which should be compared to the findings in this study as a direct comparison.*

Response 7:

Thank you for your kind suggestion.

We studied the relevant work (<https://doi.org/10.3389/fmicb.2018.01793>) and learned much from it. In the reported work, after evolution of a Δ *sdhCB* *E. coli*, mutations in α -ketoglutarate dehydrogenase (AKGDH) were found that reduced tricarboxylic acid metabolism flux and succinate secretion. Our work also found that tricarboxylic acid metabolic flux changed in TCA cycle-defected strains after evolution. Mutations in citrate synthase appeared to confer a fitness advantage and reduced α -ketoglutarate secretion. These results hinted that a glucose-grown TCA cycle-defected *E. coli* would tend to minimize carbon waste via reducing the remaining tricarboxylic acid metabolism during evolution.

According to the reviewer's suggestion, we added sentences comparing the findings in relevant work with this work in the “Discussed”: “Citrate synthase catalyzes the critical irreversible step that controls the amount of the acetyl-CoA pool and the metabolic flux into TCA cycle. In this study, mutations in citrate synthase reducing the enzyme activity of citrate synthase were found beneficial for the growth recovery of TCA cycle-deficient (Δ *aceA* Δ *sucA*) *E. coli* (Figure 1i, Supplementary Table 2, Supplementary Table 5). Tricarboxylic acid metabolic flux was decreased in evolved strain dTCA-E1, comparing to unevolved iTCA strain (Figure 2a). The mutation in the citrate synthase of strain dTCA-E1 was confirmed able to reduce α -ketoglutarate secretion of TCA cycle-deficient (Δ *aceA* Δ *sucA*) *E. coli* (Figure 4f, Supplementary Figure 5e). A previous work reported that after evolution of the *sdhCB* knock-out *E. coli*, mutations in α -ketoglutarate dehydrogenase were found that reduced tricarboxylic acid metabolic flux and succinate secretion. According to the founding of previous reported work and this study, it was assumed that a glucose-grown TCA cycle-defective *E. coli* strain would tend to minimize carbon waste by reducing the citrate (tricarboxylic acid) metabolism during evolution.” (line 422-434)

Minor Comments

Comment 1: *The canonical product of TCA is NADH and not NADPH. Please modify suitably.*

Response 1:

Sentences “In addition to providing key precursors for biomass synthesis, the oxidation of acetyl-CoA to CO₂ generates energy (ATP) and reductants (NADPH) needed for growth.” was revised to “In addition to providing key precursors for biomass synthesis, the conventional TCA cycle completely oxidizes carbon sources such as glucose to CO₂ and H₂O, generates reductants (NAD(P)H, FADH), and produces energy (ATP) primarily through coupling with oxidative phosphorylation.”(line 42).

Comment 2: [60] Ensure space between *E. coli*

Response2:

It had been corrected to “*E. coli*”. We had carefully checked the text and made all necessary corrections.

Comment 3: [128-132] In Figures 1e and 1f in the x-axis legend *SdhA* and *GltA* enzyme activity is mentioned; the assay measures the enzyme activity of the entire complex, so please change it to succinate dehydrogenase and citrate synthase enzyme activity.

Response3:

In revised Figure 1, the x-axis legend *SdhA* and *GltA* enzyme activity in previous Figure 1e-f were changed to succinate dehydrogenase and citrate synthase enzyme activity in Figure 1f-g. (line 122)

Figure 1. Enzyme activities of (f) mutated succinate dehydrogenase and (g) mutated citrate synthase in evolved strains. Error bars represent at least three independent biological replicate experiments. Ordinary one-way ANOVA multiple comparisons test was used for significant difference analysis. *: $P \leq 0.05$; **: $P \leq 0.01$; ***: $P \leq 0.001$; ****: $P \leq 0.0001$ ns: No significant difference. iTCA: unevolved BW25113 Δ *poxB::acs*; dTCA-E1, dTCA-E2, dTCA-E3, dTCA-E4: evolved dTCA.

Comment 4: [138-149] In Figure 1g and 1h, *gltA* should be italicized.

Response 4:

In revised Figure 1h-i (previous Figure 1g-h), *gltA* is italicized. (line 122)

Figure 1. (h - i) Aerobic growth of unevolved TCA cycle-defective *E. coli* dTCA after introducing succinate dehydrogenase and citrate synthase mutations. Error bars represent at least three independent biological replicate experiments. dTCA: unevolved BW25113 Δ aceA Δ sucA Δ gadA Δ gadB Δ poxB::acs.

Comment 5: [177] Figure 2c y-axis spelling of ‘balance’ to be corrected.

Response 5:

Figure 2c y-axis spelling of ‘balance’ has been corrected. (line 189)

Comment 6: Throughout the manuscript, glucose minimum medium and glucose minimal medium have been used interchangeably. Uniform nomenclature would be better.

Response 6:

Glucose minimal medium is used in the revised manuscript. All of the phrases “glucose minimum medium” had been changed to “glucose minimal medium”.

Additionally, units like mM, $g_{DCW} \cdot g_{glc}^{-1}$, $mmol \cdot g_{DCW}^{-1} \cdot h^{-1}$, $mmol \cdot g_{DCW}^{-1}$, $mol \cdot mol^{-1}$ were replaced by mmol/L, g_{DCW}/g_{glc} , mmol/ g_{DCW}/h , mmol/ g_{DCW} , mol/mol in revised manuscripts.

We had thoroughly check the text and made all necessary corrections.

Comment 7: Figures 2b and 2e are not referred to in the text.

Response 7:

In the revised manuscript, Figure 2b is referred in the added sentence: “In the dTCA-E1 strain, only 2.2% of carbon was converted into CO₂ via the TCA cycle, whereas in the iTCA strain, this percentage was much higher at 11.7% (Figure 2b, Supplementary Data 5).”(line 185)

In the revised manuscript, Figure 2e is referred in the added sentence: “The TCA cycle of dTCA-E1 neither directly generate ATP (Figure 2d) nor produce NADH/FADH (Figure 2e) for coupling with oxidative phosphorylation to produce ATP.”(line 213)

Comment 8: Main figure and corresponding replicates in the supplementary are normalized differently.

Response 8:

In the “Supplementary Materials”, Supplementary Figure 2 a-d showed the absolute carbon/cofactors metabolism rates (mmol/g_{DCW}/h) at the same biomass while with different glucose uptake rate.

Figure 2b (C mol/C mol) and Figures 2c-e (mol/100 mol glucose) were normalized to show the carbon/cofactors metabolism efficiency at the same carbon uptake rate.

Thank you sincerely for your invaluable comments and suggestions.

Responses to the reviewers' comments

To Reviewer #2:

Comment 1: Growth of the $\Delta sucA \Delta aceA$ strains. In the introduction, the authors state that “inactivation of inactivation of α -ketoglutarate dehydrogenase ($\Delta sucA$) and glyoxylate shunt ($\Delta aceA$) 1-3, 5 resulted in no aerobic growth on glucose.” (L44). However, from my reading of cited articles, it is not clearly reported that the $\Delta sucA \Delta aceA$ strain does not grow in that condition. Furthermore, adaptive laboratory evolution (ALE) was performed on a $BW25113 \Delta sucA \Delta aceA \Delta gadA \Delta gadB \Delta poxB::acs$ “to restore aerobic growth in glucose minimal medium” (L84). That strain is supposed to be not able to grow in such condition. How it is feasible to perform such an experiment if the strain is not growing? This should be clarified all along the manuscript.

Response 1:

Thank you for your comment. Smirnov *et al.* had reported that the 2Δ strain ($\Delta sucA \Delta aceA$) could not grow on glucose without any supplements (Figure 4 in the reference “Smirnov, S.V. *et al.* Metabolic engineering of *Escherichia coli* to produce (2S, 3R, 4S)-4-hydroxyisoleucine. *Appl Microbiol Biotechnol* 88, 719-726 (2010)”).

In the work reported by Theodosiou *et al.*, $3\Delta sucA$ (genotype: $\Delta aceA \Delta sucA \Delta putA$) strain could not grow on glucose without proline-4-hydroxylase reaction (Figure 2 and table 2 in reference “Theodosiou, E. *et al.* An artificial TCA cycle selects for efficient alpha-ketoglutarate dependent hydroxylase catalysis in engineered *Escherichia coli*. *Biotechnol Bioeng* 114, 1511-1520 (2017)”).

In Zhang *et al.*'s work, Figure 2b showed that 3Δ W3110 strain ($\Delta aceA \Delta sucA \Delta putA$) was unable to grow on glucose without proline-4-hydroxylase reaction (Zhang, H.L. *et al.* Efficient production of trans-4-Hydroxy-L-proline from glucose by metabolic engineering of recombinant *Escherichia coli*. *Lett Appl Microbiol* 66, 400-408 (2018)).

The reported studies showed that the *E. coli* $\Delta sucA \Delta aceA$ strain did not grow in glucose minimal medium. We had cited these references in text (line 49) and the references were listed in the “References 1-3”.

Adaptive laboratory evolution (ALE) was performed on the dTCA strain ($BW25113 \Delta sucA \Delta aceA \Delta gadA \Delta gadB \Delta poxB::acs$). As strain dTCA couldn't grow on glucose minimal medium, it was pre-cultivated in Luria-Bertani medium and subsequently transferred to glucose minimal medium for passages. As we described in the “Methods” part: “The TCA cycle-deficient strains pre-cultured overnight in Luria-Bertani medium were transferred to glucose minimal medium. For the first and second transfers, 5 mL culture was transferred into 50 mL minimal medium supplemented with 5 mL and 0.5 mL Luria-Bertani respectively. For the following serial passages, 0.5 mL culture was transferred into 50 mL glucose minimal

medium without supplements.”(line 476).

According to the reviewer’s suggestion, we have added sentences to clarify it in the manuscript: “The TCA cycle-deficient strain was pre-cultivated overnight in Luria-Bertani medium. For the first and second transfers, 5 mL culture was transferred into 50 mL glucose minimal medium supplemented with 5 mL and 0.5 mL Luria-Bertani respectively. For the following serial passages, 0.5 mL culture was transferred into 50 mL glucose minimal medium without supplements.” (line 96)

Mutations in succinate dehydrogenase was found in the survived dTCA clones from both ALE-1 and ALE-2 groups after streaking them on glucose M9 medium plate, indicating that strain dTCA have mutated during the first and second transfers (Supplementary Data 6).

Comment 2: *Succinyl-CoA supply. Authors hypothesized that the “the causal bottleneck for growth of TCA cycle-deficient E. coli was the biosynthesis of lysine and methionine due to insufficient succinyl-CoA supply”. They circumvented successfully this issue by adding in the medium lysine and methionine, or constructing strains that do not require succinyl-CoA for the biosynthesis of lysine and methionine. However, this remains indirect evidence for succinyl-CoA supply. Quantification of succinyl-CoA pool in the WT strain and in growing deficient-TCA cycle strains should provide the evidence that the latter strains are able to grow despite of a low (or inexistent) level of succinyl-CoA in the cells.*

Response 2:

Thank you for your comment.

We measured the concentration of succinyl-CoA in the WT strain (BW25113) and the deficient-TCA cycle strain (BW25113 Δ aceA Δ sucA) grown in glucose minimal medium supplemented with 1 mmol/L of methionine, DAP, and lysine. The results revealed that the WT strain had a succinyl-CoA concentration of 499.3 ± 162.5 nmol/g_{DCW}, whereas the BW25113 Δ aceA Δ sucA strain showed a significantly decreased succinyl-CoA concentration of 208.2 ± 37.8 nmol/g_{DCW}. Succinyl-CoA is the precursor for the biosynthesis of methionine, DAP, and lysine. Therefore, we hypothesized that the causal bottleneck for growth of TCA cycle-deficient *E. coli* was the biosynthesis of methionine, DAP and lysine due to insufficient succinyl-CoA supply. By supplementing methionine, DAP, and lysine, the TCA cycle-deficient *E. coli* was able to grow despite the low level of succinyl-CoA in the cells. The succinyl-CoA levels in strains DDPYM and DDPYM Δ aceA Δ sucA were determined to be 1.8 ± 0.7 nmol/g_{DCW} and 1.0 ± 0.1 nmol/g_{DCW}, respectively. However, the succinyl-CoA level in strain DDPYM Δ aceA Δ sucA Δ BCD (strain ZH40) was too low to be accurately detected.

According to the suggestion, Figure 3d was added to show the intracellular concentration of succinyl-CoA in the strains. (line 285)

Figure 3d. **(d)** Intracellular succinyl-CoA level in TCA cycle-deficient *E. coli* strains comparing to strain BW25113. 3AAs: 1 mmol/L for each of meso-2,6-diaminoheptanedioate (meso-DAP), lysine and methionine. Error bars represent at least three independent biological replicate experiments. Ordinary one-way ANOVA multiple comparisons test was used for significant difference analysis. *: $P \leq 0.05$.

The following sentences was added in the revised manuscript to show the result of succinyl-CoA level in the strains: “Additionally, the intracellular succinyl-CoA level in glucose-grown BW25113 Δ aceA Δ sucA strain supplemented with amino acid mixture (1 mmol/L each of meso-2, 6-diaminoheptanedioate, lysine and methionine) was measured as 208.2±37.8 nmol/g_{DCW}, which was significantly lower than glucose-grown BW25113 (499.3±162.5 nmol/g_{DCW}) (Figure 3d, the intracellular succinyl-CoA levels were in Supplementary Data 7).” (line 252).

“The levels of succinyl-CoA in strains DDPYM and DDPYM Δ aceA Δ sucA were determined to be 1.8±0.7 nmol/g_{DCW} and 1.0±0.1 nmol/g_{DCW}, respectively (Figure 3d, Supplementary Data 7). While, the succinyl-CoA level in strain DDPYM Δ aceA Δ sucABCD (strain ZH40) was too low to be accurately determined, with two out of the three biological replicate tests being unsuccessful (Figure 3d, Supplementary Data 7).” (line276)

The assay method for succinyl-CoA quantification was added in “Methods” as follow: “Strains were cultivated in 20 mmol/L glucose M9 minimal medium to measure the intracellular succinyl-CoA level. Then, at the log phase of growth, the strains were collected by centrifugation at 12,000 × g for 5 min. Subsequently, the collected strains were suspended (OD₆₀₀ = 20) in 1 mL of 80% (vol/vol) methanol (-80°C) for complete ultrasonication at 4°C. The resulting supernatant was collected and dried without heat in SpeedVac, while the pellet was redissolved in deionized water. The succinyl-CoA concentration was determined in the redissolved pellet using a reported LC-MS/MS method.” (line 529-536)

Comment3: Use of Δ sucABCD strain. *“2.5 TCA cycle-deficient E. coli as an efficient chassis strain for aerobic fermentations”* The authors decided to move toward the use of Δ aceA Δ sucABCD chassis instead of pursuing the work with Δ aceA Δ sucA chassis. In the former chassis, two (consecutive) TCA cycle enzymes are inactivated instead of one. There is no clear explanation for this chassis change and no data with this strain in this study is provided earlier in the manuscript. Particularly, if we guess that succinyl-CoA is also limiting

for lysine and methionine biosynthesis in such a strain, the impact of *sucCD* deletion in addition to that of *sucA* was not investigated. There are potential other effects of this double deletion on metabolism.

Response 3:

Thank you for your kind comment.

In the result “Growth of TCA cycle-deficient *E. coli* is recovered by eliminating the requirement for succinyl-CoA in amino acid pathways”. The constructed *E. coli* strain was able to grow with a severed TCA cycle ($\Delta aceA\Delta sucA$ or $\Delta aceA\Delta sucABCD$) by introducing heterogeneous acetyl-CoA dependent pathways for biosynthesis of methionine, DAP, and lysine (Figure 3c). DDPYM strain was expected to bypass the requirement of succinyl-CoA for amino acid biosynthesis (methionine, DAP, and lysine). Therefore, instead of using DDPYM $\Delta aceA\Delta sucA$, strain ZH40 (DDPYM $\Delta aceA\Delta sucABCD$) in which succinyl-CoA synthesis pathways in TCA cycle was completely blocked was utilized as a chassis to further prove the concept that the DDPYM strain could grow and perform well without both succinyl-CoA synthesis pathways (*sucAB* and *sucCD*).

The succinyl-CoA levels in strains DDPYM, DDPYM $\Delta aceA\Delta sucA$, and DDPYM $\Delta aceA\Delta sucABCD$ were measured. The results showed that the succinyl-CoA concentration in strain DDPYM was 1.8 ± 0.7 nmol/g_{DCW}, while in strain DDPYM $\Delta aceA\Delta sucA$, it was 1.0 ± 0.1 nmol/g_{DCW}. However, the succinyl-CoA level in strain DDPYM $\Delta aceA\Delta sucABCD$ (strain ZH40) was too low to accurately determine. Despite the low level of succinyl-CoA in the cells, both DDPYM $\Delta aceA\Delta sucA$ and DDPYM $\Delta aceA\Delta sucABCD$ were able to grow well (Figure 3c) and had similar specific growth rates (0.51 ± 0.05 h⁻¹ and 0.54 ± 0.06 h⁻¹, respectively) (Supplementary Table 4). Additionally, the protein expression level of DDPYM $\Delta aceA\Delta sucABCD$ was similar to that of BW25113 or DDPYM (Figure 4b). Therefore, the additional deletion of *sucABCD* in DDPYM had no significant difference compared to the deletion of *sucA* in DDPYM in terms of cell growth and protein expression. These results confirmed that in strain DDPYM $\Delta aceA\Delta sucABCD$ succinyl-CoA was no longer limiting for lysine, DAP and methionine biosynthesis. The other potential effects of the double deletion (such as *sucCD* deletion) on metabolism were not studied in this research and would be addressed in future studies.

According to the suggestion, the results of succinyl-CoA level determination were added in Figure 3d. (line 286)

Figure 3d. **(d)** Intracellular succinyl-CoA level in TCA cycle-deficient *E. coli* strains comparing to strain BW25113. 3AAs: 1 mmol/L for each of meso-2,6-diaminoheptanedioate (meso-DAP), lysine and methionine. Error bars represent at least three independent biological replicate experiments. Ordinary one-way ANOVA multiple comparisons test was used for significant difference analysis. *: $P \leq 0.05$.

The following sentences were added in the revised manuscript to show the result of succinyl-CoA level determination:

“Additionally, the intracellular succinyl-CoA level in glucose-grown BW25113 Δ aceA Δ sucA strain supplemented with amino acid mixture (1 mmol/L each of meso-2, 6-diaminoheptanedioate, lysine and methionine) was measured as 208.2±37.8 nmol/g_{DCW}, which was significantly lower than glucose-grown BW25113 (499.3±162.5 nmol/g_{DCW}) (Figure 3d, the intracellular succinyl-CoA levels were in Supplementary Data 7).” (line 252).

“The levels of succinyl-CoA in strains DDPYM and DDPYM Δ aceA Δ sucA were determined to be 1.8±0.7 nmol/g_{DCW} and 1.0±0.1 nmol/g_{DCW}, respectively (Figure 3d, Supplementary Data 7). While, the succinyl-CoA level in strain DDPYM Δ aceA Δ sucABCD (strain ZH40) was too low to be accurately determined, with two out of the three biological replicate tests being unsuccessful (Figure 3d, Supplementary Data 7).” (line 276)

The assay method for succinyl-CoA quantification was added in “Methods” as follow: “Strains were cultivated in 20 mmol/L glucose M9 minimal medium to measure the intracellular succinyl-CoA level. Then, at the log phase of growth, the strains were collected by centrifugation at 12,000 × g for 5 min. Subsequently, the collected strains were suspended (OD₆₀₀ = 20) in 1 mL of 80% (vol/vol) methanol (-80°C) for complete ultrasonication at 4°C. The resulting supernatant was collected and dried without heat in SpeedVac, while the pellet was redissolved in deionized water. The succinyl-CoA concentration was determined in the redissolved pellet using a reported LC-MS/MS method.” (line 529-536)

According to the comment, we had revised a sentence to explain why DDPYM Δ aceA Δ sucABCD was used as chassis. Sentence “To demonstrate that TCA cycle-deficient *E. coli* can serve as a universal and efficient chassis strain for aerobic fermentations, strain ZH40 (DDPYM Δ aceA Δ sucABCD) was constructed and engineered for biosynthesis of several chemicals (Figure 4a).” was revised as “To demonstrate that TCA cycle-deficient DDPYM can serve as a versatile and efficient chassis strain for aerobic fermentations without

requirement of succinyl-CoA for amino acid (methionine, DAP and lysine) biosynthesis, strain ZH40 (DDPYM $\Delta aceA \Delta sucABCD$) without succinyl-CoA synthesis pathways (*sucAB* and *sucCD*) was constructed and engineered for biosynthesis of several chemicals (Figure 4a).” (line 298)

Minor Comments

Comment 1: L21 – “improved biosynthesis yields.” Of what?

Response 1:

“Theoretically, eliminating the TCA cycle would decrease carbon dissipation and facilitate improved biosynthesis yields.” was revised to “Theoretically, eliminating TCA cycle would decrease carbon dissipation and facilitate chemicals biosynthesis.” (line 22)

Comment 2: L22 – “universal chassis for biosynthesis.” Of what?

Response 2:

“Here, we constructed an *E. coli* strain without a functional TCA cycle that can serve as a universal chassis for biosynthesis.” was revised to “Here, we constructed an *E. coli* strain without a functional TCA cycle that can serve as a versatile chassis for chemicals biosynthesis.” (line 24)

Comment 3: L39 - “the oxidation of acetyl-CoA to CO₂ generates (ATP) and reductants (NADPH) ...”: This generates mainly reductants as NADPH (1 mole) and especially FADH₂ (1 mole) and NADH (2 moles). The coupling with oxidative phosphorylation in aerobic condition provide indeed much more ATP moles than the unique mole of ATP generated in the TCA cycle.

Response 3:

The sentence “In addition to providing key precursors for biomass synthesis, the oxidization of acetyl-CoA to CO₂ generates energy (ATP) and reductants (NADPH) needed for growth.” was revised to “In addition to providing key precursors for biomass synthesis, the conventional TCA cycle completely oxidizes carbon sources such as glucose to CO₂ and H₂O, generates reductants (NAD(P)H, FADH), and produces energy (ATP) primarily through coupling with oxidative phosphorylation. The TCA cycle oxidizes each acetyl-CoA to two CO₂. The emission of CO₂ from the TCA cycle reduces carbon fluxes in the central carbon metabolism, thus negatively impacting product yield.” (line 42)

Comment 4: L135 – “was replaced with the identified mutants.” Mutant genes?

Response 4:

“To further test this hypothesis, *sdhA* was deleted in the unevolved TCA cycle-deficient

starting strain dTCA and *gltA* was replaced with the identified mutants.” was revised to “To further test this hypothesis, *sdhA* was deleted in the unevolved TCA cycle-deficient starting strain dTCA and *gltA* was replaced with the identified *gltA* mutants (GltA(T109P), GltA(L10I, T109P), GltA(H157Y) and GltA(I375T))”(line 155)

Comment 5: L147 – “*iTCA* strain, which has a complete TCA cycle”. *iTCA* is BW25113 Δ *poxB::acs*. Why using that strain and not the WT strain?

Response 5:

According to the suggestion, we had explain this in the revised manuscript: “In order to monitor the acetate (acetyl-CoA) pool and determine if the TCA cycle in the dTCA strain remained blocked during evolution, the pyruvate oxidase gene *poxB* was replaced with the acetyl-CoA synthetase gene *acs* (Δ *poxB::acs*). This replacement ensured that acetate could only be formed through acetyl-CoA and efficiently converted back to the acetyl-CoA pool.” (line 104).

The only difference between the iTCA strain and the dTCA strain was whether the genes involved in TCA cycle were deleted. Therefore, we used iTCA (BW25113 Δ *poxB::acs*) as the control for ¹³C-metabolism flux analysis, rather than BW25113.

Comment 6: L192 – “lead to ATP shortages”. Reference?

Response 6:

The statement is not rigorous. The sentence was revised. “These results challenge the common believe that TCA cycle is vital for energy generation in *E. coli* and that its defect may lead to ATP shortages. Instead, our data suggest that strain dTCA-E1 could efficiently rebalance and even enhance ATP metabolism by coordinated remodeling of overall central carbon metabolism.” were revised to “These results suggested that the TCA cycle was not important for energy generation in dTCA-E1, in which ATP metabolism was efficiently rebalanced through coordinated remodeling of central carbon metabolism.” (line 219).

Comment 7: L213 - meso-2,6-diaminoheptanedioate: Does the strain grow without adding that compound in the mixture? If no, why? If yes, why supplementing the medium with it?

Response 7:

In this study, we hypothesized that the biosynthesis of methionine, DAP and lysine in TCA cycle-deficient *E. coli* was hindered due to insufficient succinyl-CoA supply. Consequently, *E. coli* with deletion of Δ *aceA* and Δ *sucA* were unable to grow on glucose minimal medium without meso-2,6-diaminoheptanedioate (DAP), which is essential for peptidoglycan formation in Gram-negative bacteria. To investigate this, we cultivated the TCA cycle-deficient *E. coli* strain BW25113 Δ *aceA Δ *sucA* in glucose minimal medium*

supplemented with DAP, L-lysine, and L-methionine (1 mol/L for each). Remarkably, the results showed that the strain grew well under these conditions (Figure 3a, Supplementary Table 4).

According to the reviewer's suggestion, we had revised the relevant sentences:

(1) "Succinyl-CoA is involved in the biosynthesis of lysine and methionine and participates in tetrapyrroles and heme synthesis." was revised to "Succinyl-CoA is involved in the biosynthesis of methionine, DAP and lysine, and participates in tetrapyrroles and heme synthesis." (line 227)

(2) "As heme and tetrapyrroles can be synthesized from glutamyl-tRNA instead of succinyl-CoA in *E.coli*, we hypothesized that the causal bottleneck for growth of TCA cycle-deficient *E.coli* was the biosynthesis of lysine and methionine due to insufficient succinyl-CoA supply." was revised to "As heme and tetrapyrroles can be synthesized from glutamyl-tRNA instead of succinyl-CoA in *E. coli*, we hypothesized that the causal bottleneck for growth of TCA cycle-deficient *E. coli* was the biosynthesis of methionine, DAP and lysine due to insufficient succinyl-CoA supply." (line 242)

(3) "Some bacteria have evolved alternative metabolic pathways for the biosynthesis of lysine and methionine that do not require succinyl-CoA (Figure S2)" was revised to "Some bacteria have evolved alternative metabolic pathways for the biosynthesis of methionine, DAP and lysine that do not require succinyl-CoA (Supplementary Figure 3)." (line 258)

(4) "It was discovered that the limiting factor for growth was not energy limitation but insufficient supply of succinyl-CoA resulting in the inability to produce two amino acids, lysine and methionine." was corrected to "It was discovered that the limiting factor for growth was not energy limitation but insufficient supply of succinyl-CoA resulting in the inability to produce three amino acids, methionine, DAP and lysine." (line 392)

Comment 8: L255 – "However, that strain could not grow in glucose minimal medium." Not clearly mention in the cited reference. It looks like the production of deacetoxycephalosporin C is indeed toxic.

Response 8:

According to the reviewer's suggestion, we revised the relevant sentences in the introduction:

Sentences "In previous work, we blocked the TCA cycle in *E. coli* (Δ *sucA* Δ *aceA*) and reconstituted it with α -ketoglutarate-dependent deacetoxycephalosporin C synthase (DAOCS), successfully increasing the yield of G-7-aminodeacetoxycephalosporanic acid (G-7-ADCA) from penicillin G. However, since both the substrate and products of DAOCS were antibiotics and toxic to *E. coli*, we could not cultivate this engineered strain in glucose minimal medium." were revised to "In previous work, a whole-cell biocatalytic process was developed

that employed an engineered TCA cycle-deficient (*ΔsucAAaceA*) *E. coli* strain expressing α -ketoglutarate-dependent deacetoxycephalosporin C synthase (DAOCS) to convert penicillin G to G-7-ADCA. Initially, the TCA cycle-deficient strain was cultured with a rich medium and induced for overexpression of deacetoxycephalosporin C synthase, without the presence of penicillin G, during the cell growth stage. Subsequently, the resting cells, which served as the biocatalyst, were used to produce deacetoxycephalosporin C with penicillin G functioning as the substrate. However, due to the toxic nature of both the substrate and products of DAOCS on *E. coli*, it was impossible to repair the TCA cycle using the DAOCS reaction while cultivating this engineered strain in a glucose minimal medium.” (line 73-82).

Sentence “However, that strain could not grow in glucose minimal medium.” was revised to “As penicillin G and G-7-ADCA was toxic to *E. coli*, the defective TCA cycle could not be repaired with α -ketoglutarate-dependent DAOCS to support cell growth in glucose minimal medium.” (line 303)

Comment 9: L298 – *What is the benefit of using a TCA cycle-deficient strain? There is no comparison with counterpart strains having a complete TCA cycle.*

Response 9:

Blocking the TCA cycle alone did not significantly enhance the yield of acetoin due to overflow of acetate. When acetate pathway was further blocked, the yield of acetoin increased accordingly (Figure 4f, Table S5 in the previous manuscript).

In this study, we did not try to demonstrate the benefits of TCA cycle deletion on acetoin production. We used an acetoin producing strain (strain ZH42-alsSD) to trace the carbon flux from glucose to metabolites and demonstrate the benefit of TCA cycle deficiency in minimizing carbon loss. (Figure 4f, Supplementary Figure 5f, Supplementary Data 8). In the revised manuscript, we had deleted the results of the acetoin production (Figure 4f and Figure S4c; part of Table S5 in the previous manuscript, and the related text (line 280-311)).

We introduced the *Ptac-alsSD* expression cassette into TCA cycle intact strains BW25113 and DDPYM, resulting in strain BW25113-alsSD and DDPYM-alsSD. Both BW25113-alsSD and DDPYM-alsSD were unable to grow aerobically in the glucose M9 minimal medium, unlike ZH42-alsSD. We assumed that the competition between acetoin synthesis and the intact TCA cycle, might cause excessive consumption of pyruvate which resulted in growth arrest. Therefore, a comparative analysis with counterpart strains possessing a complete TCA cycle could not be done.

Comment 10: L328 – *“It has long been believed that a complete TCA cycle is necessary for aerobic respiration and energy production in heterotrophic microbe.” Adding a reference would be better.*

Response 10:

The statement is not rigorous, and we have deleted it in the revised manuscript.

Comment 11: Figure 1: The specific glucose consumption rates would be also of interest.

Response 11:

The specific glucose consumption (uptake) rates have been added in Figure 1d and referred in text as “The biomass yields of all evolutionary endpoint strains were significantly lower than that of the iTCA strain (Figure 1c, Supplementary Data 1), while the glucose uptake rates of those four evolved strains were significantly higher than that of iTCA strain (Figure 1d, Supplementary Data 1).” (line 119)

Figure 1. (d) Glucose uptake rates of evolved *E. coli* TCA cycle-deficient strains. Error bars represent at least three independent biological replicate experiments. Ordinary one-way ANOVA multiple comparisons test was used for significant difference analysis. *: $P \leq 0.05$; **: $P \leq 0.01$; ***: $P \leq 0.001$; ****: $P \leq 0.0001$; ns: No significant difference. dTCA: unevolved BW25113 Δ sucA Δ aceA Δ gadA Δ gadB Δ poxB::acs; iTCA: unevolved BW25113 Δ poxB::acs; dTCA-E1, dTCA-E2, dTCA-E3, dTCA-E4: evolved dTCA.

Comment 12: Figure 2: The choice of colors may complicate the analysis of the figure for colorblind readers (same for supplementary figures).

Response 12:

According to the reviewer's suggestion, the colors of Figure 2 and Supplementary Figure 2 had been modified.

Thank you sincerely for your invaluable comments and suggestions.

Responses to the reviewers' comments

To Reviewer #3:

Comment 1: Rational for deleting the TCA cycle. The authors say in the introduction that “it is believed that blocking the TCA cycle and its bypass pathways could reduce carbon losses and improve product yields in aerobic fermentations.” This is not really clear, because the TCA cycle is much more efficient in generating energy than fermentation. The authors should clearly explain why deletion of the TCA cycle has benefits. I understand that the yield improves in non- or slow-growing strains but a growth arrest can be achieved by deletion of many enzymes and other cellular processes. What is the exact mechanism that makes the dTCA strain a better host?

Response 1:

Thank you for your kind suggestion.

In this study, we constructed an *E. coli* strain without a functional TCA cycle that can serve as a versatile chassis for chemicals biosynthesis. The elimination of the conventional TCA cycle prevents carbon dissipation from acetyl-CoA complete oxidization, thus facilitating chemicals production. For heterotrophs, the conventional TCA cycle facilitates the complete oxidation of carbon sources like glucose to CO₂, providing reductants (NAD(P)H, FADH) and ATP to support cell growth. Each acetyl-CoA is oxidized, releasing two CO₂ molecules (one from decarboxylation of isocitrate and one from decarboxylation of α -ketoglutarate). However, the emission of CO₂ from the TCA cycle reduces the carbon flux of the central carbon metabolism, negatively impacting product yield. In the whole-cell catalysis solution without nitrogen or phosphorus, strains BW25113 and DDPYM, which had an intact TCA cycle, showed complete oxidation of glucose despite halted cell growth (Figure 4e-f, Supplementary Figure 5a-b). On the other hand, in the TCA cycle defected strains (ZH40, ZH44, ZH45, ZH42-alsSD), glucose mainly underwent conversion into metabolites like acetate and AKG (Figure 4e-f, Supplementary Figure 5c-f).

According to the reviewer's suggestion, we had made revision to clarify it in the revised manuscript. Sentences “In addition to providing key precursors for biomass synthesis, the oxidization of acetyl-CoA to CO₂ generates energy (ATP) and reductants (NADPH) needed for growth. However, it is believed that blocking the TCA cycle and its bypass pathways could reduce carbon losses and improve product yields in aerobic fermentations.” were revised to “In addition to providing key precursors for biomass synthesis, the conventional TCA cycle completely oxidizes carbon sources such as glucose to CO₂ and H₂O, generates reductants (NAD(P)H, FADH), and produces energy (ATP) primarily through coupling with oxidative phosphorylation. The TCA cycle oxidizes each acetyl-CoA to two CO₂. The emission of CO₂ from the TCA cycle reduces carbon fluxes in the central carbon metabolism,

thus negatively impacting product yield. Therefore, blocking the TCA cycle and its bypass pathways could decrease carbon dissipation and facilitate chemical biosynthesis in aerobic fermentations.” (line 43).

Comment 2: Evolution experiment. The starting strain does not grow (in Figure 1a dTCA has a growth rate of 0), but the authors say that it needs 230 generations for evolution. How does a non-growing strain divide 230 times?

Response 2:

Thank you for your comment.

Adaptative laboratory evolution (ALE) was performed on the dTCA strain (*BW25113ΔsucAΔaceAΔgadAΔgadBΔpoxB::acs*). As strain dTCA couldn't grow on glucose minimal medium, it was pre-cultivated in Luria-Bertani medium and subsequently transferred to glucose minimal medium for passages. As we described in the “Methods” part: “The TCA cycle-deficient strains pre-cultured overnight in Luria-Bertani medium were transferred to glucose minimal medium. For the first and second transfers, 5 mL culture was transferred into 50 mL minimal medium supplemented with 5 mL and 0.5 mL Luria-Bertani respectively. For the following serial passages, 0.5 mL culture was transferred into 50 mL glucose minimal medium without supplements.”(line 475).

According to the reviewer’s suggestion, we have added sentences to clarify it in the manuscript: “The TCA cycle-deficient strain was pre-cultivated overnight in Luria-Bertani medium. For the first and second transfers, 5 mL culture was transferred into 50 mL glucose minimal medium supplemented with 5 mL and 0.5 mL Luria-Bertani respectively. For the following serial passages, 0.5 mL culture was transferred into 50 mL glucose minimal medium without supplements.” (line 96)

Mutations in succinate dehydrogenase was found in the survived dTCA clones from both ALE-1 and ALE-2 groups after streaking them on glucose M9 medium plate, indicating that strain dTCA have mutated during the first and second transfers (Supplementary Data 6).

*Comment 3: Whole genome sequencing (WGS). The authors performed WGS to identify the ALE mutations. However the results are not clearly presented and only mutations in *sdhA* and *gltA* are presented. Were these the only mutations? The authors should attempt a more systematic analysis to understand if there are adaptive mutations in other pathways.*

Response 3:

Thank you for your comment.

In addition to the mentioned TCA cycle gene mutations, numerous other mutations were identified. Details of all these mutations can be found in Supplementary Data 2. Although we focused on TCA cycle genes mutations which was commonly mutated in all of four evolved

dTCA strains in this study, researchers could seek for mutations in their interested pathway from Supplementary Data 2.

Sentence “Several genes coding for enzymes in the TCA cycle were found to be mutated in the evolved strains” was revised to “The analysis of gene mutations in the evolved strains revealed common mutations in genes encoding enzymes in the TCA cycle (Figure 1e, Supplementary Data 2).” (line 142)

According to the reviewer’s suggestion, we added the detail of genome sequencing and mutations identification in the method: “The genome DNA of the evolved strains was extracted and prepared as Illumina PE150 libraries. It was then sequenced using Illumina HiSeq X technology by MajorBio company. The BWA-Samtools-Bcftools SNP calling pipeline was employed to identify mutations, with the *E. coli* BW25113 genome (Accession: CP009273.1) serving as the reference sequence. Only variations that met the criteria of being covered by more than 50 reads and located in the ORF or the upstream 100 bp untranslated region were retained. The genome of unevolved dTCA was also sequenced and analyzed to eliminate any innate genetic variations before ALE experiment. The reads associated with this study have been deposited in the National Center for Biotechnology Information (NCBI) databank SRA, with the BioProject accession number (Accession: PRJNA967749). SNP analysis for the four evolutionary endpoint strains were listed in Supplementary Data 2.”(line 486-496)

Comment 4: *Interpretation of the Flux Analysis. In the flux map of the dTCA strain it is unclear which reaction produces succinate. The authors should also explain if all of the fluxes are balanced and how they included biomass production in the model.*

Response 4:

Thank you for your comment.

In the dTCA-E1 strain, succinate might be produced from fumarate by (1) L-aspartate oxidase (NadB) or (2) fumarate reductase.

L-aspartate oxidase (NadB) is the first enzyme in the pathway catalyzing the FAD-dependent oxidation of L-aspartate to iminoaspartate. NadB can use either oxygen or fumarate as oxidant. When fumarate is used, it is reduced to succinate. Fumarate is a conventional substrate for NadB and is kinetically superior to oxygen (Korshunov and Imlay, 2010. Two sources of endogenous hydrogen peroxide in *Escherichia coli*. Molecular Microbiology, 2010, 75(6), 1389–1. <https://doi.org/10.1111/j.1365-2958.2010.07059.x>). The intracellular concentration of fumarate would be sufficient to provide the low concentrations of succinate required for growth of the $\Delta lipB \Delta sdhB$ strains (Fatemah A. Hermes, John E. Cronan. An NAD synthetic reaction bypasses the lipoate requirement for aerobic growth of

Escherichia coli strains blocked in succinate catabolism. *Molecular Microbiology* (2014) 94(5), 1134–1145 doi.org/10.1111/mmi.12822). In the dTCA-E1 strain, NadB might convert fumarate to succinate, which could supply low concentrations of succinate.

Another source of succinate might come from the action of fumarate reductase (FRD), which could supply limited amounts of succinate. Fumarate reductase is an iron-sulfur protein that is optimally active under anaerobic conditions (Cecchini, G., Schröder, I., Gunsalus, R. P., and Maklashina, E. (2002). Succinate dehydrogenase and fumarate reductase from *Escherichia coli*. *Biochim. Biophys. Acta* 1553, 140–157. doi: 10.1016/S0005-2728(01)00238-9). However, even when oxygen is present, FRD is able to catalyze the oxidation of succinate at a ratio of 1:1.5 compared to reduction of fumarate (Hirsch, C. A., Rasminsky, M., Davis, B. D., and Lin, E. C. (1963). A fumarate reductase in *Escherichia coli* distinct from succinate dehydrogenase. *J. Biol. Chem.* 238, 3770–3774.)

According to the reviewer’s suggestion, we had added sentences to explain the reactions that might produce succinate: “As SdhA was inactivated, succinate pool in dTCA-E1 might be supplemented from fumarate by L-aspartate oxidase (NadB) or fumarate reductase (FRD) (Supplementary Figure 1).” (line 180). Supplementary Figure 1 was added to show the potential replenishment pathway for succinate. (line 11 in the Supplementary Materials)

Supplementary Figure 1. Potential replenishment pathway for succinate through L-aspartate oxidase (NadB) and fumarate reductase (FRD) reactions

The biomass reaction in the ¹³C-MFA model was based on the biomass composition measured for wild-type *E. coli* in this paper: (Long CP, Antoniewicz MR (2019) Metabolic flux responses to deletion of 20 core enzymes reveal flexibility and limits of *E. coli* metabolism. *Metab Eng* 55: 249–257). Details of network model for flux analysis were listed in Supplementary Data 3. All fluxes are balanced in the model. This biomass reaction describes the amount of precursors and co-factors required to produce 1 gram of dry weight cell mass of *E. coli*.

Comment 5: Claim that TCA cycle is not vital for energy generation is misleading. The authors

report that ATP formation is higher in the dTCA strain, but it is not clear why this supports the claim that the TCA cycle is not vital for energy production. It only shows that this strain has a higher demand for ATP. What is the reason for the higher ATP demand? Are there futile cycles that waste ATP?

Response 5:

Thank you for your kind comment.

The statement is not rigorous. We had revised the sentence. Sentence “These results challenge the common believe that TCA cycle is vital for energy generation in *E. coli* and that its defect may lead to ATP shortages. Instead, our data suggest that strain dTCA-E1 could efficiently rebalance and even enhance ATP metabolism by coordinated remodeling of overall central carbon metabolism.” were revised to “These results suggested that the TCA cycle was not important for energy generation in dTCA-E1, in which ATP metabolism was efficiently rebalanced through coordinated remodeling of central carbon metabolism.” (line 219)

Besides, we had added a sentence to explain why the TCA cycle in dTCA-E1 was not important for energy generation in the revised manuscript: “The TCA cycle of dTCA-E1 neither directly generate ATP (Figure 2d) nor produce NADH/FADH (Figure 2e) for coupling with oxidative phosphorylation to produce ATP.” (line 213)

We speculated that there might be a futile cycle that consumes ATP, such as the cycle between acetate and acetyl-CoA. The futile cycle resulted in the higher ATP metabolism efficiency in the dTCA-E1 strain. Acetate can be produced through the PoxB or the AckA-Pta pathways (Biotechnol. Prog. 21:1062-1067, 2005; Biotechnol. Lett. 32:1897-1903, 2010). In the dTCA-E1 strain, the pyruvate oxidase gene *poxB* was replaced with the acetyl-CoA synthetase gene *acs* ($\Delta poxB::acs$) to regenerate acetyl-CoA from acetate. AMP-forming acetyl-CoA synthetase (ACS) catalyzes the formation of acetyl-CoA from acetate, which consumes ATP and produces AMP. Thus, the conversion between acetate and acetyl-CoA might be one of the reasons for the higher ATP demand in dTCA-E1.

Comment 6: *Replacing SucCoA dependent pathways. The motivation for replacing the SucCoA enzymes is not clear. First the authors evolved the dTCA strain and found that deleting sdhA in the sucA background improves the strain. Do I understand it correctly that now the goal is to use a second approach to improve the dTCA strain by rational design that resulted in the DDPYM strain? This part should be better connected with the evolution part, especially because the DDPYM strain is used as production host (so why doing the evolution in the first place).*

Response 6:

Thank you for your kind comment.

Through a systematic adaptive laboratory evolution (ALE) study, we firstly conducted the evolution process to obtain a well-grown TCA cycle deficient *E. coli* strain. The purpose

of this was to analyze the growth recovery mechanism of a TCA cycle defective *E. coli* in a glucose minimal medium. Our findings revealed that the sole limiting factor for growth was the shortage of succinyl-CoA (SucCoA). Additionally, we discovered that the mutation of succinate dehydrogenase was necessary and inevitable for relieving the TCA cycle defect in the *E. coli* strain ($\Delta aceA \Delta sucA$) (Supplementary Data 6). However, the mutation of succinate dehydrogenase would be ineffective for producing chemicals, such as G-7-ADCA, that are generated through α -ketoglutarate (AKG) dependent enzyme reactions. This method requires the regeneration of AKG from succinate via the TCA cycle with succinate dehydrogenase. To create a versatile chassis for producing different chemicals, we conducted experiments that eliminated the need for SucCoA and prevented succinate dehydrogenase mutations.

According to the reviewer's suggestion, the following sentences was added in the revised manuscript to better connect the evolution part with the rational design of the DDPYM strain. “Succinate dehydrogenase mutations were found in all 30 randomly isolated clones (mutations analysis of citrate synthase and succinate dehydrogenase were listed in Supplementary Data 6) when we sequenced the succinate dehydrogenase operon in dTCA or BW25113 $\Delta aceA \Delta sucA$ clones that survived on glucose M9 minimal medium plate. This finding indicated that succinate dehydrogenase mutation was seemingly inevitable. However, *E. coli* with defective succinate dehydrogenase can't be used to produce chemicals like G-7-ADCA, as it requires the TCA cycle to regenerate α -ketoglutarate from succinate. Hence, we explored an alternative strategy to circumvent the succinyl-CoA requirement.” (line 233-240)

The revised manuscript discussed the reason for a strategy change to circumvent succinyl-CoA after conducting initial evolution. Following sentences was added:

“Adaptive laboratory evolution was used to gain a better understanding of the underlying mechanisms causing the growth defect observed when TCA cycle was severed at the α -ketoglutarate dehydrogenase step in *E. coli*. It was discovered that the limiting factor for growth was not energy limitation but insufficient supply of succinyl-CoA resulting in the inability to produce three amino acids, methionine, DAP and lysine. Succinyl-CoA can only be generated from succinate in *E. coli* if the α -ketoglutarate dehydrogenase reaction was blocked ($\Delta sucA$). When the glyoxylate cycle was also blocked by inactivation of isocitrate lyase ($\Delta aceA$), the *E. coli* with defective TCA-cycle ($\Delta aceA \Delta sucA$) lacked sufficient succinate to maintain the succinyl-CoA pool. Although succinate might be supplemented from fumarate by L-aspartate oxidase (NadB) or fumarate reductase (FRD) (Supplementary Figure 1), the amount of succinate was limited. Due to succinate being rapidly oxidized by succinate dehydrogenase, TCA cycle defected ($\Delta aceA \Delta sucA$) *E. coli* could not maintain a sufficient succinyl-CoA pool without enough intracellular succinate, unless its succinate dehydrogenase was mutated. Upon transfer to glucose minimal medium plates from rich medium, those

survived dTCA and BW25113 *ΔaceAΔsucA* clones were found to have succinate dehydrogenase mutations (Supplementary Data 6). This work provides evidence that succinate dehydrogenase mutation is necessary and unavoidable for recovering aerobic growth of TCA-cycle defective *E. coli* (*ΔaceAΔsucA*) in glucose minimal medium.

However, succinate dehydrogenase mutation was inapplicable to the production of chemicals such as G-7-ADCA, which required a reconstituted TCA cycle to regenerate α -ketoglutarate from succinate. Therefore, we attempted to circumvent succinyl-CoA and construct a more versatile chassis. It is a challenge to bypass succinyl-CoA requirement, for succinyl-CoA has always been considered as one of the twelve essential precursor metabolites for cell growth. This work showed that growth could be recovered by bypassing the requirement for succinyl-CoA in amino acids synthesis through expression of heterologous acetyl-CoA dependent enzymes. Succinyl-CoA synthesis pathways genes *sucAB* and *sucCD* were knocked out in TCA cycle-deficient *E. coli* strain ZH40, which still retained the ability to provide essential precursor metabolites and energy for aerobic growth (Figure 3c, Supplementary Figure 4b) and over expression of protein (Figure 4b) while avoiding unnecessary oxidization of acetyl-CoA to CO₂ in the TCA cycle (Figure 4e-f, Supplementary Figure 5c, Supplementary Data 8). The succinyl-CoA level in ZH40 was hundreds of times lower than that of wild type *E. coli* BW25113, making it too low to determine accurately (Figure 3d, Supplementary Data 7).” (line 389-421)

Comment 7: Overproduction of chemicals. The authors show mostly titers (g/L), but to show that flux into the product increases they should report fluxes in g/L per g Biomass. Overall the section is difficult to understand because different genetic modifications are made for the different products and it is unclear which modification results in the improved production rates. I suggest that the authors highlight better the advantage of the TCA deletion, e.g. higher substrate levels at the AKG branch point.

Response 7:

Thank you for your kind suggestion. In the revised manuscript, acetate production per g biomass (mmol/g_{DCW}) during cell growth was added in Supplementary Table 5 (line 63 in Supplementary Materials). The method for calculating production per g biomass was added as: “The production per gram of dry cell weight (mmol/g_{DCW}) was determined using linear regression on metabolite concentrations and biomass dry weight during log phase of growth” (line 520). The production of glutamate per g biomass (mmol/g_{DCW}) was not reported due to the failure of the linear regression on glutamate concentrations and biomass during the growth log phase. This was consistent with the fact that glutamate mainly accumulated after the growth log phase (Figure 4d). Figure 4c and Supplementary Figure 5 shown the production of chemicals by resting cells during whole-cell catalysis. The titers (mmol/L) in these figures, with the same amount of biomass (3.2 g_{DCW}/L) and feedstock, could illustrate the advantages

of TCA cycle deficiency for chemical production. Figure 4d-e shows the accumulation of glutamate and acetate along with time during aerobic growth. Therefore, the volumetric production (mmol/L) was still used in Figure 4c-e and Supplementary Figure 5.

To illustrate the benefits of TCA deletion in chemical production, the strains were modified to achieve high yields of various products, such as G-7-ADCA, glutamate, and acetate. In ZH40 chassis (DDPYM $\Delta aceA\Delta sucABCD$), yields of all these chemicals were increased with a deficient TCA cycle.

TCA deletion in strain ZH40 resulted in a 5-fold increase in G-7-ADCA production (7.0 ± 0.3 mmol/L) compared to the complete TCA cycle strain DDPYM strain (1.4 ± 0.05 mmol/L) (Figure 4c).

Strain ZH40-gdhA-ppc produced 5 mmol/L glutamate (Figure 4d), while strains with an intact TCA cycle (BW25113-gdhA-ppc or DDPYM-gdhA-ppc) had no detectable glutamate production (Figure 4d). Furthermore, AKG accumulation in strain ZH40 during whole-cell biocatalysis was much higher than that in BW25113 and DDPYM (Supplementary Figure 5). These experiments provided confirmation of the advantageous effect of TCA deletion on chemical production from AKG.

For acetate production, ZH40 (DDPYM $\Delta aceA\Delta sucABCD$) and ZH44 (DDPYM $\Delta aceA\Delta sucABCD\Delta poxB$) produced acetate at a higher yield and rate than *E. coli* strains with an intact TCA cycle (Figure 4e, Supplementary Table 5), while maintaining a similar growth rate as BW25113 and DDPYM (Supplementary Figure 4b, Supplementary Table 5). In the whole-cell catalysis, acetate yield from glucose was 0.96 ± 0.008 mol/mol (strain ZH40 or ZH44) and 1.42 ± 0.04 mol/mol (ZH45) in TCA-cycle defective DDPYM (Supplementary Figure 5c-e, Supplementary Data 8), while no acetate was detected in TCA cycle intact strains (BW25113 or DDPYM) (Supplementary Figure 5a-b, Supplementary Data 8). These experiments confirmed the benefit of TCA deletion for chemicals production from AcCoA.

In the whole-cell biocatalysis by the TCA-cycle defective strains, such as ZH40, ZH44, and ZH45, the major carbon flux from glucose to metabolites like acetate and AKG was measured. In strain ZH42-alsSD, where the acetoin synthesis pathway was introduced, glucose was nearly completely converted to metabolites with a tiny amount of carbon loss (Figure 4f). On the other hand, in BW25113 and DDPYM, all carbon was almost lost when glucose was consumed (Figure 4f). These results demonstrated decreased carbon dissipation of TCA cycle-deficient strain.

We had modified Figure 4, Supplementary Figure 4 and Supplementary Table 5 (Figure 4f and Figure S3c and part of Table S5 in the previous manuscript was removed). Acetoin synthesis was used to track the carbon flux from glucose to metabolites, and the fraction of carbon flux from glucose to metabolites during whole-cell catalysis was shown in revised Figure 4f (line 369).

Figure 4. High yield biosynthesis of chemicals using TCA cycle-deficient *E. coli* strains. **(a)** Overview of products produced using engineered ZH40 strains. **(b)** Overexpression of deacetoxycephalosporin C synthase (DAOS) in the whole-cell biocatalysis and **(c)** production of G-7-ADCA in the TCA cycle-deficient ZH40 strain. **(d)** Production of glutamate in TCA cycle-deficient chassis ZH40 with plasmid pSC2s-gdhA-ppc. **(e)** Production of acetate in TCA cycle-deficient chassis strains comparing to BW25113 and DDPYM. **(f)** Fraction of carbon flux from glucose to metabolites during whole-cell catalysis. Ppc: phosphoenolpyruvate carboxylase; GdhA: glutamate dehydrogenase; AlsS: alpha-acetolactate synthase; AlsD: alpha-acetolactate decarboxylase; AckA: acetate kinase A; Pta: phosphate acetyltransferase; DAOS: deacetoxycephalosporin C synthase; DDPYM: BW25113 $\Delta dapD::dapH(Bs)-dapL(Bs)-patA(Bs)$ $\Delta metA::ycjI(Bs)-metA(Bs)$. Detailed information on all strains is listed in Supplementary Table 1. Error bars represent at least three independent biological replicate experiments. Ordinary one-way ANOVA multiple comparisons test was used for significant difference analysis. ****: $P \leq 0.0001$.

We also add Supplementary Figure 5 (line 37 in Supplementary Materials) to show the advantage of the TCA deletion on supply of AKG (AKG accumulation) and AcCoA (acetate accumulation):

Supplementary Figure 5. The whole-cell biocatalysis by the strains (a) BW25113, (b) DDPYM, (c) ZH40, (d) ZH44, (e) ZH45, (f) ZH42-alsSD, using glucose as substrate. DDPYM: BW25113 $\Delta dapD::dapH(Bs)-dapL(Bs)-patA(Bs) \Delta metA::ycjI(Bs)-metA(Bs)$; ZH40: DDPYM $\Delta aceA\Delta sucABCD$; ZH44: ZH40 $\Delta poxB$; ZH45: ZH44 $gltA(T109P)$; ZH42: ZH40 $\Delta poxB\Delta pta gltA(T109P)$; ZH42-alsSD: ZH42 $\Delta ldhA::Ptac-alsSD$. Detailed information of all strains was listed in Table S1. Error bars represent at least three independent biological replicate experiments.

According to the reviewer’s suggestion, we added the following sentences to show the advantage of the TCA deletion on chemical production with higher substrate levels at the AKG branch point: “In the whole-cell biocatalysis, little amount of AKG were detected in BW25113 and DDPYM (Supplementary Figure 5a-b), while strain ZH40 (DDPYM $\Delta aceA\Delta sucABCD$) with defective TCA cycle produced 14.9 ± 0.51 mmol/L AKG after complete consumption of 50 mmol/L glucose (Supplementary Figure 5c). These results demonstrated the advantage of the defective TCA cycle on AKG accumulation and the TCA-cycle defective strain held potential in production of chemicals derived from AKG.” (line 316-321).

According to the reviewer’s suggestion, sentences “During whole-cell biocatalysis, acetate yield increased further to 0.96 ± 0.01 mol/mol in ZH44 (Table S6), and 1.42 ± 0.04 mol/mol in ZH45 (Table S6), while no acetate was accumulated in cultures of *E. coli* strains with an intact TCA cycle (Table S6).” were revised to “During whole-cell biocatalysis, acetate yield increased further to 0.96 ± 0.01 mol/mol in ZH40 and ZH44 (Supplementary Figure 5c-d,

Supplementary Data 8), and 1.42 ± 0.04 mol/mol in ZH45 (Supplementary Figure 5e, Supplementary Data 8), while no acetate was accumulated in cultures of *E. coli* strains with an intact TCA cycle (Supplementary Figure 5a-b, Supplementary Data 8). These results demonstrated the benefit of the defective TCA cycle for production of chemicals from acetyl-CoA.”(line 332-337)

According to the reviewer's suggestion, we have carefully revised sentences regarding the tracking of carbon flux from glucose to metabolites and further explained the benefit of TCA deletion in achieving efficient chemical production while minimizing carbon waste. The sentences about acetoin synthesis were revised. The sentences (line 280 - line 311) in the previous manuscript was deleted and revised as following in the revised manuscript:

“In whole-cell catalysis stage, the resting cells did not grow. The yields of metabolites from glucose in whole-cell catalysis stage (Supplementary Figure 5) were used to analyze the distribution of metabolic flow (fractions of carbon fluxes from glucose to metabolites were listed in Supplementary Data 8). To calculate the fraction of carbon fluxes from glucose to metabolites (Figure 4f), the measured yields of metabolites were divided by the theoretical yields of their respective pathways, with CO₂ and any unmeasured metabolites considered as lost carbon. In the strains with complete TCA cycle, BW25113 and DDPYM, the TCA cycle decarboxylation caused complete carbon loss from glucose (Figure 4f, Supplementary Figure 5a-b, Supplementary Data 8). However, the TCA cycle-deficient strains ZH40, ZH44, and ZH45 showed a significant reduction in carbon loss, with over 70% of the carbon flux diverted to acetate and α -ketoglutarate (AKG) (Figure 4f, Supplementary Figure 5c-e, Supplementary Data 8). Mutation of citrate synthase in strain ZH45 leads to a decrease in AKG flux and an increase in acetate flux compared to strain ZH44 (Figure 4f, Supplementary Figure 5 d-e, Supplementary Data 8). However, these TCA cycle defective strains ZH40, ZH44, and ZH45 also presented above 20% carbon loss. Citrate synthase mutation caused a modest increase in lost carbon in strain ZH45 (26.54 ± 2.17 %) comparing to strain ZH44 (20.44 ± 0.92 %) (Figure 4f, Supplementary Data 8). We hypothesized that there might exist carbon loss from acetyl-CoA or pyruvate in strain ZH45, which might be decreased if carbon flux was directed from pyruvate to pyruvate-derived chemical such as acetoin. Acetoin can be synthesized from pyruvate using acetolactate synthase (AlsS) and acetolactate decarboxylase (AlsD) without production or consumption of reductants or energy (Figure 4a). Genes *alsS* and *alsD* from *Bacillus subtilis* were introduced into strain ZH42, resulting in strain ZH42-*alsSD* (Supplementary Table 1). In whole-cell catalysis, strain ZH42-*alsSD* generated 35.98 ± 0.57 mmol/L acetoin from 50 mmol/L glucose, which was 72% of the theoretical maximal yield from glucose (Supplementary Figure 5f, Supplementary Data 8). Only 0.17 ± 0.67 % of the carbon in ZH42-*alsSD* was lost from glucose, and most of carbon flowed to metabolites including pyruvate (4.14 ± 0.94 %), acetoin (71.95 ± 1.13 %), acetate

(11.65±0.87 %), and α -ketoglutarate (12.09±0.35 %) (Figure 4f, Supplementary Data 8). These results demonstrated the significant potential of TCA cycle-deficient DDPYM strains for minimizing carbon dissipation and facilitating chemicals production.” (line 339-368)

Comment 8: The authors don't provide insights about the mechanisms that improved production in the TCA deficient strain. This is reflected in a very brief discussion, at least here the authors could provide hypothesis about why a TCA deletion is beneficial for overproduction.

Response 8:

Thank you for your kind comment.

The revised discussion delves into the mechanisms that led to improved production in the TCA deficient strain. Following sentences were add:

“Citrate synthase catalyzes the critical irreversible step that controls the amount of the acetyl-CoA pool and the metabolic flux into TCA cycle. In this study, mutations in citrate synthase reducing the enzyme activity of citrate synthase were found beneficial for the growth recovery of TCA cycle-deficient (*$\Delta aceA \Delta sucA$*) *E. coli* (Figure 1i, Supplementary Table 2, Supplementary Table 5). Tricarboxylic acid metabolic flux was decreased in evolved strain dTCA-E1, comparing to unevolved iTCA strain (Figure 2a). The mutation in the citrate synthase of strain dTCA-E1 was confirmed able to reduce α -ketoglutarate secretion of TCA cycle-deficient (*$\Delta aceA \Delta sucA$*) *E. coli* (Figure 4f, Supplementary Figure 5e). A previous work reported that after evolution of the *sdhCB* knock-out *E. coli*, mutations in α -ketoglutarate dehydrogenase were found that reduced tricarboxylic acid metabolic flux and succinate secretion. According to the founding of previous reported work and this study, it was assumed that a glucose-grown TCA cycle-defective *E. coli* strain would tend to minimize carbon waste by reducing the citrate (tricarboxylic acid) metabolism during evolution.

For a heterotrophic microbe like *E. coli*, glucose is entirely oxidized to CO₂ and H₂O through the TCA cycle and respiratory chain in aerobic conditions. However, complete oxidation of each acetyl-CoA via the TCA cycle leads to the emission of two CO₂, which reduces the carbon flux directed to product formation and negatively affects product yield. Therefore, eliminating the TCA cycle in *E. coli* can decrease carbon dissipation and improve the efficiency of chemicals biosynthesis. To prove this concept, the TCA cycle-deficient DDPYM strain was developed and successfully used to achieve higher yield biosynthesis of chemicals from acetyl-CoA (acetate) and AKG point (G-7-ADCA, glutamate) when compared to strains with an intact TCA cycle (Figure 4, Supplementary Figure 5). On the other hand, *E. coli* strains with an intact TCA cycle couldn't prevent carbon loss from acetyl-CoA oxidation, even when their growth is strictly restricted in a whole-cell catalysis solution (Figure 4e-f, Supplementary Figure 5a-b, Supplementary Data 8). The chassis strain

ZH40, with the genotype DDPYM $\Delta aceA \Delta sucABCD$, accumulated α -ketoglutarate (Figure 4f, Supplementary Figure 5c, Supplementary Data 8), making it suitable for the efficient biosynthesis of various products that require α -ketoglutarate, such as G-7-ADCA, hydroxy-proline, hydroxy-isoleucine, and glutamate family amino acids. On the other hand, the chassis strain ZH42, with the genotype DDPYM $\Delta aceA \Delta sucABCD \Delta poxB \Delta pta$ *gltA(T109P)*, is a versatile chassis for producing chemicals derived from acetyl-CoA (e.g., butanol and fatty acids) as well as chemicals derived from glycolysis pathway intermediates (e.g., pyruvate). In summary, this work opens up exciting new avenues for more efficient biosynthesis of industrial chemicals in the future.” (line 422-455)

Thank you sincerely for your invaluable comments and suggestions.

Reviewers' Comments:

Reviewer #1:

Remarks to the Author:

The authors have responded to my comments adequately. Nothing further based on the changes.

Reviewer #2:

Remarks to the Author:

Overall, the authors provide a thorough response to my initial review with significant improvement.

However, I have still some comments (Line number according to the tracking version):

L44 – “the conventional TCA cycle completely oxidizes carbon sources such as glucose to CO₂ and H₂O”. H₂O results from O₂ reduction and not from carbon oxidation.

L45 – “FADH” does not exist. Please replace by FADH₂ (Here and everywhere in the manuscript and figures/tables).

L46 – “The emission of CO₂ from the TCA cycle reduces carbon fluxes in the central carbon metabolism”. This sentence is not clear. The CO₂ production results from carbon fluxes in the central metabolism. How does it reduce the carbon fluxes? I guess this is not the meaning of the authors (Yield instead of flux?). This should be rephrased

L79 – Δ sucA Δ aceA instead of Δ aceA Δ sucA.

L113 – “... and efficiently converted back to the acetyl-CoA pool”. Not sure it is the case due to posttranslational and metabolic regulations on Acs. Pta-AckA pathway is likely the main route of acetate assimilation in this situation (Enjalbert et al., 2017, Scientific reports).

L183 – “..., while citrate synthase attenuation although not essential was beneficial”. Complicated to read. I suggest, “while citrate synthase attenuation, although not essential, was beneficial”.

L202 – “FRD”. Please use/uniformize the nomenclature for protein names (Frd – Here and everywhere in the manuscript and figures/tables).

L228 – “whereas in iTCA the transhydrogenase flux was a major contributor to biosynthetic NADPH production”. How does this flux is calculated? Should be mentioned in M&M.

L303 – “While, the succinyl-CoA level in strain DDPYM Δ aceA Δ sucABCD (strain ZH40) was too low to be accurately determined, with two out of the three biological replicate tests being unsuccessful (Figure 3d, Supplementary Data 7).” Something goes wrong in the sentence.

L359 – “0.64±0.004 h⁻¹”. Problem with significant figures (same remark L360).

Reviewer #3:

Remarks to the Author:

I thank the authors for a thorough revision and for addressing all of my points. This is now a much stronger manuscript. I have no further points.

The manuscript “ A citric acid cycle-deficient *Escherichia coli* as an efficient chassis for aerobic fermentations”(NCOMMS-23-15866B) has been carefully revised. We appreciate the kind instructions from the editor and the invaluable and detailed comments and suggestions from the reviewers. The point-by-point responses to the reviewers’ comments are listed below.

List of actions (main changes made in the manuscript)

LOA1: “FADH” in figures was corrected as “FADH₂” (Figure 2e and Supplementary Figure 2d).

LOA2: “FRD” in Supplementary Figure 1 was corrected as “Frd”.

LOA3: Method for calculating transhydrogenase flux was added (line 631-640). The cited reference was added in the “References” part (line 789).

LOA4: According to reviewer’s comments, we revised the significant figures in text (line 326-327) and Supplementary Table 5.

LOA5: According to reviewer’s comments, we revised relevant sentences (line 42, line 43, line 45, line 75, line 108, line 153, line 173, line 208, line 274, line 326-327, line 400, line 830).

LOA6: Subheadings of “Results” was rewritten in a more concise manner to meet the requirements (no longer than 60 characters) according to the “Guide to formatting articles” of Nature Communications.

LOA7: According to requirements of the “Author Checklist”, part of data in Supplementary Data files were moved to Source Data file, so that, supplementary data files 1-9 were reduced to Supplementary Data files 1-7. The format of supplementary data were revised (one excel document contain one worksheet according to “Author Checklist”). Relevant citations in text were also revised.

LOA8: According to requirements of the “Author Checklist”, reference of Source Data, definition of error bars, exact values of ‘n’ and *p* values are added, titles or legends of some Figures/Tables, the Data availability section, the Acknowledgements section, and the Competing interests section are revised.

Responses to the reviewers' comments

To Reviewer #2:

Comment 1:

L44 – “the conventional TCA cycle completely oxidizes carbon sources such as glucose to CO₂ and H₂O”. H₂O results from O₂ reduction and not from carbon oxidation.

Response 1:

Thank you for your kind comments. According to the reviewer's suggestion, the sentence “.....the conventional TCA cycle completely oxidizes carbon sources such as glucose to CO₂ and H₂O.....” was revised to “.....the conventional TCA cycle completely oxidizes carbon sources such as glucose to CO₂” (line 42)

Comment 2:

L45 – “FADH” does not exist. Please replace by FADH₂ (Here and everywhere in the manuscript and figures/tables).

Response 2:

FADH was replaced by FADH₂ in the text (line 43, line 208, line 830), Figure 2e and Supplementary Materials (Supplementary Figure 2d, line 18).

Comment 3:

L46 – “The emission of CO₂ from the TCA cycle reduces carbon fluxes in the central carbon metabolism”. This sentence is not clear. The CO₂ production results from carbon fluxes in the central metabolism. How does it reduce the carbon fluxes? I guess this is not the meaning of the authors (Yield instead of flux?). This should be rephrased

Response 3:

Thank you for your kind comments. Sentence “The emission of CO₂ from the TCA cycle reduces carbon fluxes in the central carbon metabolism, thus negatively impacting product yield.” revised to “The emission of CO₂ from the TCA cycle reduces carbon fluxes directed to products synthesis, thus negatively impacting product yield.” (line 45)

Comment 4:

L79 – Δ sucA Δ aceA instead of Δ aceA Δ sucA.

Response 4:

“ Δ sucA Δ aceA” was replaced by “ Δ aceA Δ sucA”. (line 75)

Comment 5:

L113 – “.... and efficiently converted back to the acetyl-CoA pool”. Not sure it is the case due to posttranslational and metabolic regulations on Acs. Pta-AckA pathway is likely the main route of acetate assimilation in this situation (Enjalbert *et al.*, 2017, *Scientific reports*).

Response 5:

Thank you for your kind comments.

We agree with the reviewer that Pta-AckA pathway serves as the primary route for acetate assimilation in wild type *E. coli* (Enjalbert *et al.*, 2017, *Scientific reports*). Acetate can be metabolized by two alternative pathways to form acetyl-CoA: the reversible Pta-AckA pathway, or the irreversible acetyl-CoA synthetase. The induction of Acs expression occurs either because of cell growth on acetate (T D Brown *et al.*, The enzymic interconversion of acetate and acetyl-coenzyme A in *Escherichia coli*. *J Gen Microbiol.* 102: 327-336 (1977)), or due to mutations in the *acs* promoter region (D S Treves, S Manning, J Adams. Repeated evolution of an acetate-crossfeeding polymorphism in long-term populations of *Escherichia coli*. *Mol Biol Evol.* 15: 789-797 (1998)). Upon acetylation by acetyl-phosphate, the acetyl-CoA synthetase activity of Acs is diminished, amounting to approximately 60% of that exhibited by wild type Acs. (Fengying Liu *et al.*, Acs is essential for propionate utilization in *Escherichia coli*. *Biochem Biophys Res Commun.* 449: 272-277 (2014)). Nevertheless, in our study, Acs was overexpression (an additional copy of *acs* was incorporated into the *poxB* locus for overexpression). On this basis, Acs is expected to be functional in this situation. The activity of Acs, in concert with the constitutive activity of Pta and AckA, contributed to acetate assimilation in this situation. We did not measure and compare the activities of Acs and Pta/AckA, so we are unable to determine which pathway was the main route for acetate assimilation in the strain.

According to the reviewer’s suggestion, we had revised the sentence “This replacement ensured that acetate could only be formed through acetyl-CoA and efficiently converted back to the acetyl-CoA pool.” as follow: “In this situation, acetate formed from acetyl-CoA was converted back to the acetyl-CoA pool.” (line 107)

Comment 6:

L183 – “...., while citrate synthase attenuation although not essential was beneficial”.

Complicated to read. I suggest, “while citrate synthase attenuation, although not essential, was beneficial”.

Response 6:

Thank you for your kind comments. Sentence “These results suggested that succinate dehydrogenase inactivation was essential for the growth recovery of TCA cycle defected *E. coli*, while citrate synthase attenuation although not essential was beneficial.” was revised to “These results suggested that succinate dehydrogenase inactivation was essential for the growth recovery of TCA cycle defected *E. coli*, while citrate synthase attenuation, although not essential, was beneficial.” (line 153)

Comment 7:

L202 – “FRD”. Please use/uniformize the nomenclature for protein names (*Frd* – Here and everywhere in the manuscript and figures/tables).

Response 7:

“FRD” was replaced by “Frd” in text (line 173, line 400) and in Supplementary Materials (Supplementary Figure 1 and line 10).

Comment 8:

L228 – “whereas in *iTCA* the transhydrogenase flux was a major contributor to biosynthetic NADPH production”. How does this flux is calculated? Should be mentioned in M&M.

Response 8:

According to the reviewer’s suggestion, the method for determining transhydrogenase flux had been supplemented in “Methods”, as follow: “The transhydrogenase flux and oxidative phosphorylation fluxes in the model are determined using the carbon fluxes estimated from ¹³C-MFA. In short, in the metabolic model used for ¹³C-MFA, all reactions are redox balanced. The assignment whether NADH, FADH₂ or NADPH is oxidized or reduced in any reaction is based on the known biochemistry of *E. coli* enzymes. The total production and consumption fluxes for each cofactor are calculated from ¹³C-MFA results and then balanced. To balance NADPH fluxes, a transhydrogenase reaction is included in the metabolic model. To balance NADH and FADH₂ fluxes, oxidative phosphorylation reactions are included in the model. The results of flux analysis and cofactor balance analysis were listed in Supplementary Data 4.” (line 631-640) . **Comment 9:**

L303 – “While, the succinyl-CoA level in strain DDPYM *ΔaceAΔsucABCD* (strain ZH40) was too low to be accurately determined, with two out of the three biological replicate tests being

unsuccessful (Figure 3d, Supplementary Data 7).” Something goes wrong in the sentence.

Response 9:

Sentence “While, the succinyl-CoA level in strain DDPYM $\Delta aceA\Delta sucABCD$ (strain ZH40) was too low to be accurately determined, with two out of the three biological replicate tests being unsuccessful (Figure 3d, Supplementary Data 7).” was corrected as “While, the succinyl-CoA level in strain DDPYM $\Delta aceA\Delta sucABCD$ was too low to be accurately measured in biological replicate tests (Figure 3d, Supplementary Data 6)”. (line 273)

Comment 10:

L359 – “ $0.64 \pm 0.004 \text{ h}^{-1}$ ”. Problem with significant figures (same remark L360).

Response 10:

Sentence “.....while maintaining a similar growth rate ($0.63 \pm 0.01 \text{ h}^{-1}$ and $0.64 \pm 0.004 \text{ h}^{-1}$) as BW25113 ($0.66 \pm 0.01 \text{ h}^{-1}$) and DDPYM ($0.63 \pm 0.002 \text{ h}^{-1}$) (Supplementary Figure 4b, Supplementary Table 5).” was corrected as “.....while maintaining a similar growth rate ($0.63 \pm 0.01 \text{ h}^{-1}$ and $0.64 \pm 0.0040.01 \text{ h}^{-1}$, respectively) as BW25113 ($0.66 \pm 0.01 \text{ h}^{-1}$) and DDPYM ($0.63 \pm 0.002 \text{ h}^{-1}$) (Supplementary Figure 4b, Supplementary Table 5)” (line 326-327)

The same problem in Supplementary Table 5 was also corrected. (line 69 in Supplementary Materials)

Thank you sincerely for your invaluable comments and suggestions.